# Neural networks differentiate between Middle and Later Stone Age lithic assemblages in eastern Africa

**Matt Grove**[1]*, **James Blinkhorn**[2,3]

**1** Department of Archaeology, Classics and Egyptology, University of Liverpool, Liverpool, United Kingdom, **2** Pan-African Evolution Research Group, Max Planck Institute for the Science of Human History, Jena, Germany, **3** Department of Geography, Royal Holloway, University of London, Egham, Surrey, United Kingdom

☯ These authors contributed equally to this work.
* matt.grove@liverpool.ac.uk

**Data Availability Statement:** All relevant data are within the manuscript and its Supporting Information files.

**Funding:** MG and JB were supported the Natural Environment Research Council as part of Grant NE/

## Abstract

The Middle to Later Stone Age transition marks a major change in how Late Pleistocene African populations produced and used stone tool kits, but is manifest in various ways, places and times across the continent. Alongside changing patterns of raw material use and decreasing artefact sizes, changes in artefact types are commonly employed to differentiate Middle Stone Age (MSA) and Later Stone Age (LSA) assemblages. The current paper employs a quantitative analytical framework based upon the use of neural networks to examine changing constellations of technologies between MSA and LSA assemblages from eastern Africa. Network ensembles were trained to differentiate LSA assemblages from Marine Isotope Stage 3&4 MSA and Marine Isotope Stage 5 MSA assemblages based upon the presence or absence of 16 technologies. Simulations were used to extract significant indicator and contra-indicator technologies for each assemblage class. The trained network ensembles classified over 94% of assemblages correctly, and identified 7 key technologies that significantly distinguish between assemblage classes. These results clarify both temporal changes within the MSA and differences between MSA and LSA assemblages in eastern Africa.

## Introduction

The transition from the Middle Stone Age (MSA) to Later Stone Age (LSA) signals a major shift in the lithic assemblages produced by African *Homo sapiens* populations, but this transition occurs over a considerable period, is manifest in numerous ways, and occurs asynchronously in different regions of the continent. In eastern Africa, the onset of this transition begins as early as 67 thousand years ago [ka] at Panga ya Saidi [1], appearing in southern Rift Valley sites such as Enkapune ya Moto [2], Mumba [3], Magubike [4], Nasera [5] before 45ka, whilst recognised MSA technologies persist later in assemblages in the southwestern Ethiopian highlands at Fincha Habera [6] and Mochena Borago [7], in the Horn of Africa at Goda

K014560/1, "A 500,000-year environmental record from Chew Bahir, south Ethiopia: testing hypotheses of climate-driven human evolution, innovation, and dispersal", which forms part of the Hominin Sites and Paleolakes Drilling Project. https://nerc.ukri.org/ The funders had no role in study design, data collection and analysis, decision to publish, or preparation of the manuscript.

**Competing interests:** The authors have declared that no competing interests exist.

Buticha [8–10], and in the Lake Victoria Basin [11]. The transition is often characterised archaeologically by decreases in prepared-core technologies and the production of retouched points and by concomitant increases in (backed) microliths, prismatic blade(lets) and bipolar reduction methods [12]. In addition, frequencies of ochre use and bead production are also generally considered to increase across the transition from MSA to LSA, though both first appear long prior to the earliest transitional sequences. The appearance of the various innovations generally employed to characterise the LSA must thus be understood as a "time-transgressive" process [12] that differed in both chronology and technology between regions. Many of the classical lithic indicators of Middle and Later Stone Age technologies were defined in relation to the evidence from southern Africa [e.g.13–15], and whilst these have been successfully applied to the eastern African record in the past, the number of published eastern African assemblages is now sufficient to explicitly analyse the dynamics of the transition in this region within a more rigorous quantitative framework.

The pressing need for quantitative analysis stems from the fact that the terms 'LSA' and 'MSA' remain poorly defined; archaeologists tend to rely on a few key *fossiles directeurs* of each industrial complex, but the particular tools forms identified for this purpose vary considerably between researchers and formal definitions remain elusive. A recent review [12], for example, makes it clear that even the 'characteristic artefacts' associated with a given industrial complex are neither unique to that complex nor always present in assemblages identified as belonging to that complex. That a recent meeting of leading MSA researchers failed to develop a 'unified analytical approach' [16] also hints at the scale of the problem. The analyses reported below attempt to shift the debate from one focussed on individual artefact types towards the consideration of recurring associations between constellations of technologies that mark genuine, demonstrable divisions in the data.

It should be noted at the outset that the terms such as 'MSA' and 'LSA' have been extensively criticised for failing to fully capture the variation found in the African lithic record, for imposing discrete, rigid categorical boundaries on continuous phenomena, and for artificially homogenising material within chronological or technological 'blocks' [17–19]. Though these terms remain the most frequently used in describing the archaeology of the period, some [e.g. 18, 20] have reverted to the use of Clark's [21] five 'modes' as an alternative method for describing lithic variation. For Barham and Mitchell, a key advantage of the mode system is that applying one particular term (e.g. 'Mode 3') to a given assemblage "does not imply that other techniques were not also used" [18, p.16]. However, if the main problem with 'MSA' and 'LSA' is that assemblages labelled 'MSA' can sometimes contain elements more typically considered 'LSA' (and vice-versa) then replacing this terminological system with one in which assemblages labelled 'Mode 3' can sometimes contain elements more typically considered 'Mode 4' does not appreciably improve matters. Barham and Mitchell [18] still employ the names of particular industries when discussing the details of regional trajectories, but embed these into the mode system at broader geographic and temporal scales.

Shea's [19] dissatisfaction with the standard 'Stone Ages' terminology has led to the East African Stone Tool (EAST) typology [19], a novel mode and sub-mode system that focuses explicitly on strategies for modifying lithic artefacts (see also [22]). This is a potentially useful scheme, as its application involves enumerating the presence or absence of each mode in each assemblage; it therefore avoids labelling assemblages uniquely as, for example, 'MSA' or 'Mode 3'. It is likely that Shea's [19] scheme will combine well with quantitative analyses such as those reported below, but this will only be possible once each assemblage has been individually described in accordance with the EAST typology. Since the current paper is concerned with one (relatively short) chronological period in one (relatively large) region, the terms 'LSA' and 'MSA' remain appropriate. Further to this, the dataset analysed here utilises existing

classifications employed by the researchers who excavated or analysed a given assemblage; these classifications are overwhelmingly either 'MSA' or 'LSA'. It would be inherently wrong to re-assign those assemblages either to different categories of the same classificatory scheme or to categories of an alternate classificatory scheme prior to analysis.

The current paper therefore employs a broad typology [adapted from 23] (see S1 File) to examine changing constellations of technologies between Middle and Later Stone Age assemblages from eastern Africa. Given the history of research in the region, this focuses on records from Ethiopia, Kenya and Tanzania, alongside evidence from Eritrea, Somaliland and Uganda. This typology – which considers technologies based on their presence or absence in each assemblage – has enabled construction of a far larger dataset than would have been possible if absolute frequencies of individual tool forms were employed. The dataset facilitates not only comparison of the LSA with the MSA, but also consideration of changes within Late Pleistocene MSA records, split between Marine Isotope Stage (MIS) 5 (130-71ka) and MIS 4 (71-59ka) and 3 (59-29ka). This extensive dataset is here coupled with a sophisticated machine learning process that allows for the extraction of significant indicators of each industry or period. The primary advantage of this process is that it is not subject to the requirements for particular distributions in the raw data, nor to the assumption of linearity between variables that limits so many traditional parametric statistical methods. The specific aim of the analyses reported below is to identify which technologies differ significantly between MIS5, MIS3 & 4 MSA, and LSA assemblages in terms of their presence or absence in those assemblages. The analyses do not attempt to test the general efficacy of neural networks as classification algorithms, as such tests have been performed repeatedly and at length elsewhere. As noted above, the utility of the terms 'LSA' and 'MSA' has been questioned; these terms are retained here for reasons also outlined above, but it should be noted that an alternative classification based on, for example, Clark's [21] mode system, could produce alternative results. The sections below briefly outline the nature of the MSA / LSA transition in eastern Africa and introduce artificial neural networks as a classification tool for archaeological data analysis.

## The MSA/LSA transition

The transition from Middle Stone Age (MSA) to Later Stone Age (LSA) marks a substantive change in Pleistocene human behaviour. The MSA first appears in eastern Africa ~300 thousand years ago (ka) at the site of Olorgesailie, where the combination of Levallois technologies and a diverse retouched toolkit were recovered alongside utilised ochre and evidence from procurement of raw materials over considerable differences [24]. Notably, this is broadly contemporaneous with the first fossils attributed to *Homo sapiens* found in north-western Africa at Jebel Irhoud [25]. Middle Stone Age technologies can be found across the continent, persisting in eastern Africa until the end of MIS 3 [e.g. 6] Marine Isotope Stage (MIS) 3, and enduring in other regions into the terminal Pleistocene [26]. The LSA also first appears in eastern Africa ~67ka at the site of Panga ya Saidi, where significant changes in artefact size and patterns of raw material use accompany an increasing focus on bipolar technologies, alongside alternating appearances of blade and Levallois methods, and continue to be used into the Holocene [1]. The chronological overlap between the youngest MSA and oldest LSA in eastern Africa is significant, lasting up to 35 thousand years, highlighting the need for transparent means to resolve between the two in order to evaluate the factors that drive such behavioural changes.

A number of changes in lithic technology have been argued to signal major shifts in patterns of human behaviour relating to this transition, including shifts in artefact typology, raw material use and artefact sizes. Our recent quantitative appraisal of the presence and absence of artefact types in MSA assemblages from eastern Africa identified Levallois and blade

technologies, alongside discoidal cores, retouched points, scrapers and denticulates as key components of these stone tool assemblages [23]. Studies of early LSA industries in eastern Africa have similarly highlighted key typological components, including the dominant use of bipolar technology [19–20], and the appearance of prismatic blade production and backed geometric pieces [1]. The use of individual typological indicators to differentiate MSA and LSA technologies is, however, complicated by the appearance of key LSA technologies, such as bipolar technologies, blade production and backing, within MSA assemblages [23], and continuity of important MSA technologies, such as Levallois reduction, within LSA assemblages [1,5]. Tryon and colleagues [5] note that proportional, rather than categorical, changes may be important in resolving between MSA and LSA assemblages, although identifying suitable proportional thresholds may not be straightforward.

A notable decrease in artefact size has also been identified as an important trend between MSA and LSA assemblages in eastern Africa, providing an additional means to discriminate between these industries [1,27–29]. Pargeter and Shea [30] have discussed the significance of miniaturisation as a persistent trend in stone tool use through time. Notably, they highlight changes in artefact size in eastern Africa that appear decoupled from patterns in artefact typology, reinforcing the apparent typological continuities noted above. However, they present evidence for miniaturised stone tool technologies with MSA industries in eastern Africa, such as in the Lower Omo and Middle Awash valleys, as well as their prevalence in LSA industries. Nevertheless, clear shifts in artefact sizes can be identified in individual site sequences, such as at Panga ya Saidi, that are interpreted as signals of substantive technological change [1]. In addition to changes in artefact sizes, a shift in focus of raw material use has been associated with the MSA-LSA transition in eastern Africa [1,29]. In particular, there is an increasing focus on more fine grained raw material types and those that appear in smaller clast sizes, such as crystal quartz, which may accompany the decrease in artefact sizes. Again, such a shift may be best understood in terms of shifting proportions of raw material use, rather than a clear categorical shift.

Historically, the transition from MSA to LSA technologies has been explained by significant changes in cognition [31,32], identified not only through a shift in stone tool use argued to reflect the appearance of more sophisticated methods of hunting, such as the appearance of hafting and use of diverse projectile technologies [33], but also through the fluorescence of conspicuous indicators of symbolic behaviour, such as beads. With a body of evidence supporting the appearance of symbolic behaviour in MSA contexts emerging from the late 1980s [e.g. 34] and accelerating towards the turn of the century [35], modern explanations for this transition focus on regional variability in patterns of ecological adaptation, demography, and social change [36–38]. Within this context, examining which elements of stone tool technologies, or constellation of elements, appear as key features of either the MSA and LSA, and conversely those that have little clear relationship to such categories, is an important step to focus further study on the processes impacting upon cultural inheritance, technological innovation and behavioural change.

In order to engage with the largest number of MSA and LSA assemblages reported from eastern Africa, we focus on patterns of the presence and absence of stone artefact technologies reported from chronometrically dated sites in the literature. Although reports of patterns of raw material use and artefact sizes are commonplace, their means of reporting vary more considerably that artefact typologies, drastically reducing the number of assemblages available for comparison. Previous researchers have suggested that proportional, rather than qualitative, changes in artefact typologies better characterise technological changes observed across the MSA-LSA transition. However, a range of factors, from the method and intensity of archaeological research methodologies to patterns of sediment accumulation, the formation of

behavioural palimpsests and the position of a site within its geographic and ecological context, may have a substantial impact upon archaeological assemblage composition. Some [e.g. 39] would even go as far as to argue that lithic assemblages – and the individual artefacts that they contain – represent arbitrary points in a reduction continuum, and that archaeological typologies do not represent 'finished artefacts'. While such factors can be constrained more readily by detailed studies at individual sites, their impact on wider synthetic approaches may be harder to control. We acknowledge the issues relating to reduction of lithic assemblages to a catalogue of the presence or absence of different stone tool technologies or types (see S1 File), but suggest this offers the best means to provide the widest overview of change across the MSA-LSA transition, which can help to target more detailed studies in the future.

## Artificial neural networks

Artificial Neural Networks (henceforth ANNs) are computer models that are intended to mimic the salient features of information processing in the brain. Like the brain, their considerable processing power arises not from the complexity of any single unit but from the action of many simple units acting in parallel. Each node of an ANN is intended to represent a single neuron in that it sums a series of inputs to determine an output. The structure of a single artificial neuron is shown in Fig 1; this differs only slightly from the format originally proposed by McCulloch and Pitts [40,41] and later referred to as the 'perceptron' by Rosenblatt [42,43]. As a tool for data classification, the single perceptron suffers from many of the drawbacks of standard statistical techniques; notably, it can only solve linearly separable problems [44] (see Fig 2). The combination of multiple artificial neurons of this type, however, overcomes many of these problems and allows Multi-Layer Perceptrons (henceforth MLPs) to accurately represent

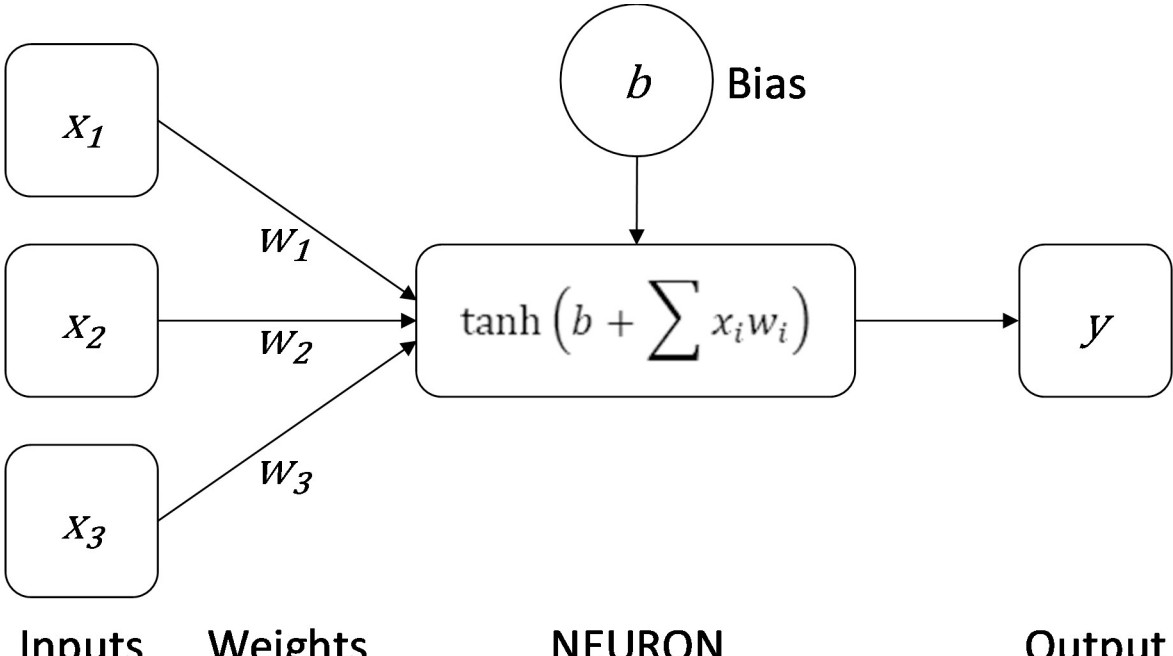

**Fig 1. The structure of a single artificial neuron.** Inputs (in this case presences or absences of technologies in an assemblage) are multiplied by 'synapse' weights; these products are then summed and added to the bias of the neuron. Finally, an activation function (in this case the hyperbolic tangent) is applied to this sum to determine the strength at which the neuron will 'fire' (i.e. the activation function determines the value that will be propagated on to the neuron(s) in the next layer of the network).

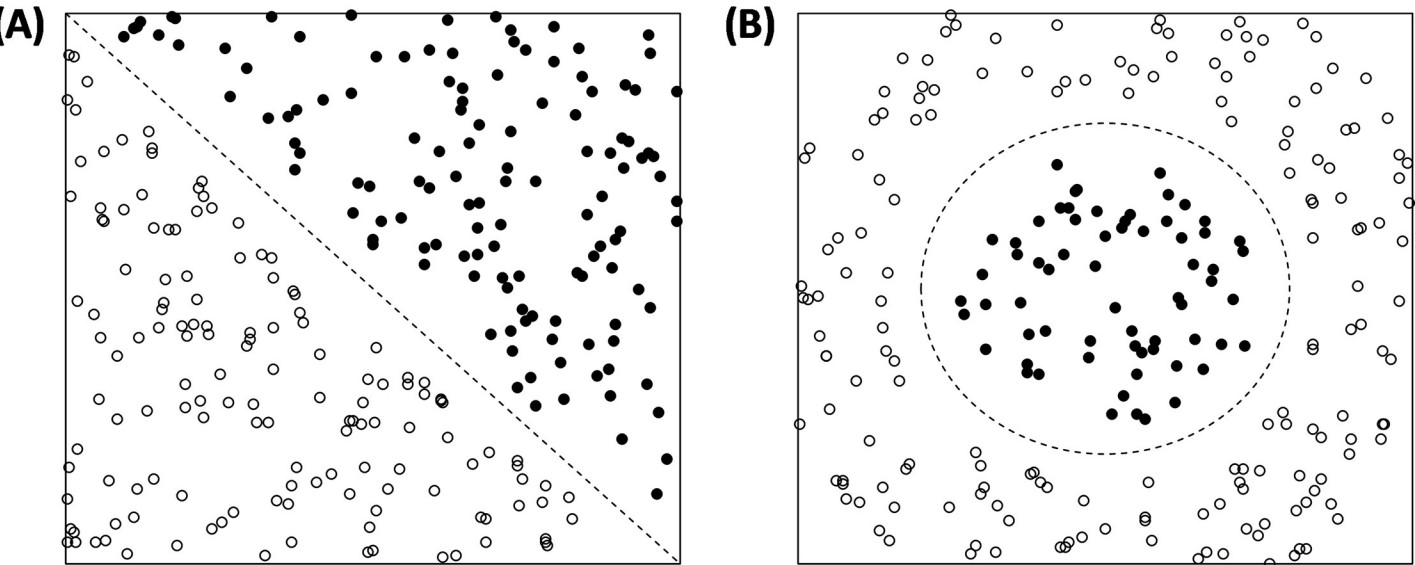

**Fig 2. Linear and non-linear separability.** (A) shows two classes of data that can be separated by a single straight line (i.e. they are 'linearly separable'). (B) shows two classes of data that require a more complex classification topology (i.e. they are not linearly separable). Neural networks are capable of solving the complex topography in (B), whereas more traditional statistical methods (such as binary logistic regression, the obvious comparator in this case) are not.

and learn complex nonlinear relations between classes. The back-propagation algorithm, popularised by Rumelhart and colleagues [45], provides a fast, efficient method for training MLPs: errors, defined as differences between the actual and the desired classifications, are fed backwards through the network in order to update the biases of individual neurons and the weights that connect them (see Fig 1).

The development of Artificial Neural Networks (henceforth ANNs) has a considerable history [40–46], yet applications of neural computing in general are very rare in archaeology [47,48]. Where such methods have been applied, they have generally been used to search for patterns in 2-dimensional image data derived via remote sensing [49,50] or during material provenance analysis [51]. With the exception of Nash and Prewitt's [52] pioneering study of Texas projectile point typology, we are aware of no other published research using ANNs to classify archaeological lithic assemblages.

This lack of engagement with ANNs is surprising, given that they offer several advantages over traditional statistical techniques that are likely to be particularly relevant to archaeological data analysis. ANNs excel at solving classification problems based on "complex, noisy, partial information" [53] of the kind that often comprises archaeological datasets. As mentioned above, ANNs can learn complex non-linear boundaries between classes. A corollary of this ability is that the forms of relationships between independent and dependent variables are not specified in advance, but instead emerge empirically during the learning process [54:354ff.]. Input data need not conform to any parametric distribution, and data of different types (continuous, ordinal, nominal) can be input simultaneously [55:2630ff.]. Automatic rescaling prior to analysis ensures that input variables demonstrating greater variance will not have disproportionate effects on the output. The sophistication of the back-propagation algorithm ensures that ANNs rarely become trapped in local minima, and the simulation and aggregation of the results of multiple networks negates this problem if and when it does occur.

Asparoukhov and Krzanowski [56] demonstrate that ANNs are particularly effective when dealing with sparse binary data of the form employed below. Multinomial logistic regression,

the traditional statistical method for analysing such a dataset, is a cumbersome process the results of which are often difficult to interpret. ANNs act as nonlinear generalizations of logistic regression in which all direct and interaction terms are modelled, and their classification accuracy is frequently greater than that of their statistical equivalent, particularly in more complex cases [54]. In a direct comparison of the two methods on a series of classification problems, Tu [57] concluded that ANNs can be used to model "much more complex nonlinear relationships than a logistic regression model". The efficacy of ANNs as classification tools has thus been demonstrated on multiple occasions [e.g. 54,56,57].

One criticism that has often been levelled at ANNs is that the mechanism of translation from inputs to outputs remains opaque; as Cross and colleagues [58:1079] observe, "data go in and predictions come out, but the user has no understanding of what happens in between". Basic ANN output does not include the kind of information normally reported following a standard statistical test and, although an equation describing the relationship between inputs and outputs can be formulated in matrix notation, it does little to aid interpretation. However, bootstrap simulations such as those described by Baxt [59,60; see below] allow for the derivation of values equivalent to effect sizes, as well as their associated 95% confidence intervals. Such simulations substantially undermine the 'black box' objection to ANNs. Since ANNs are necessarily stochastic (initial weights and biases and allocation to training, test, and validation sets are random), averaging across a large sample of networks trained to solve a given problem also provides greater robusticity and confidence in the generated results.

## Methods

### Data

The tool typology used is adapted from that of Blinkhorn & Grove [23] to enable integration of LSA assemblages. Following a synthesis of the literature of all reported artefact types, the presence or absence of 16 artefact forms was catalogued for each of the 92 eastern African MIS2-5 assemblages spanning the MSA to LSA transition including: Backed Pieces, Bipolar Technology, Blade Technology, Borer, Burin, Centripetal Technology, Core Tool, Denticulate, Levallois Blade Technology, Levallois Flake Technology, Levallois Point Technology, Notch, Platform Core, Point Technology, RT Bifacial and Scraper. These terms encompass the breadth of terminology used to describe stone tool assemblages for Late Pleistocene eastern Africa, and enable assembly of a substantial dataset. Full details regarding the choice of terminology and the assembly of the dataset are reported in SI. Whilst it is clear that some MSA elements persist beyond MIS2, it is generally considered that the main transitional phase occurs in MIS3, and the data were therefore limited to MIS2-5 to enable analysis of changes both within the MSA and between broadly contemporary MSA and LSA assemblages. This dataset is expanded relative to Blinkhorn and Grove [23] to include 31 LSA, 31 MIS3&4 MSA, and 30 MIS5 MSA assemblages (see S1 File for further details). In all cases, the designation of an assemblage as LSA or MSA follows the designation given by the original researchers. Many of the assemblages included here have been subject to recent dating studies enabling high levels of chronological resolution, but to integrate this with the breadth of evidence available and to accommodate the error ranges associated with dating, here we group assemblages that can be confidently attributed to each MIS. This enables us to examine perceived differences in behaviour evident between MSA and LSA assemblages during their extensive overlap during MIS 4 and 3, that would be overlooked in a purely chronological division of the dataset and potential obscure intra-regional variability. As with any archaeological analysis of this kind, there is a reliance upon the work of numerous previous researchers over an extended period of time during which the application of finer-grained typological designations is likely to have been

inconsistent. The aggregation of multiple different typological terms into broad categories (see S1 File for full details), the focus on presence or absence of technologies rather than frequency of tool forms, and the chronological aggregation into MI Stages are attempts to overcome this inconsistency as far as is reasonably possible. The spatial distribution of assemblages is shown in Fig 3.

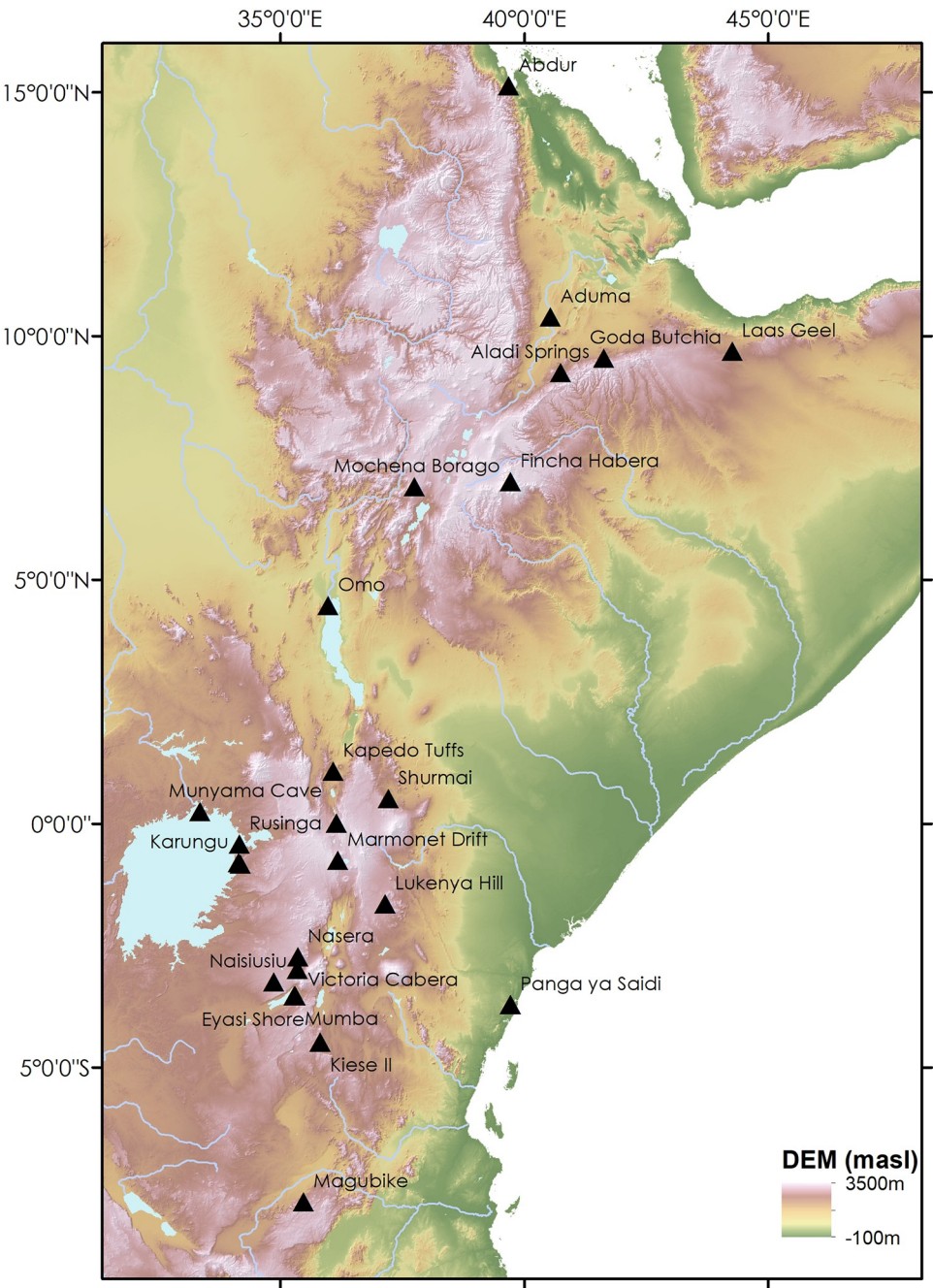

**Fig 3. Map illustrating the distribution of sites contributing assemblages to the dataset for analysis, plotted on a digital elevation model (SRTM, 1 arc-second) obtained from USGS earth explorer ([61]: https://earthexplorer. usgs.gov).**

## Assemblage classification

Two separate sets of analyses were run, dividing the data into either two (LSA, MSA) or three (LSA, MIS3&4 MSA, and MIS5 MSA) classes. This allows for simultaneous examination of both overall differences between the LSA and MSA and changes within the MSA itself. As both the initialisation weights of an ANN and the division into training and test sets are set randomly, it is considered preferable to train an ensemble of networks rather than a single network to classify data. Classification results can then be averaged over the ensemble of networks using a weighted mean with the weights proportional to the performance of each network. Accordingly, 1,000 feed-forward ANNs were trained to distinguish between LSA, MIS3&4 MSA, and MIS5 MSA assemblages, and a further 1,000 networks to distinguish between LSA and MSA assemblages. The ANNs employed each have three layers: an input layer of 16 nodes (one for each technology), a hidden layer of 10 nodes, and an output layer of two or three nodes (one for each assemblage class). The number of nodes in the hidden layer should be minimised to avoid over-fitting; initial experiments demonstrated that hidden layers with <10 nodes performed relatively poorly, whereas those with >10 nodes did not appreciably increase classification performance.

Partition into training and test subsets was carried out via randomised indices to ensure that each network used the following numbers of assemblages for each subset in the 3-way analyses:

- Training: 26 LSA, 26 MIS3&4 MSA, 25 MIS5 MSA;

- Testing: 5 LSA, 5 MIS3&4 MSA, 5 MIS5 MSA.

Equivalently, randomised indices were used to ensure the following division for each subset in the 2-way analyses:

- Training: 26 LSA, 52 MSA;

- Testing: 5 LSA, 9 MSA.

This corresponds as closely as possible to the ideal split of 85% for training and 15% for testing. These percentages were held constant throughout, with the actual assemblages appearing in each subset varying randomly between networks. The result of this random indexing procedure is that, across the 1,000 networks, each assemblage is included in the training set on an average of 850 occasions and in the testing set on an average of 150 occasions.

The networks were trained using a Bayesian Regularization (BR) algorithm [62,63], which is more robust than many of the more often used back-propagation algorithms. BR is particularly useful for relatively small samples as its internal regularization procedure precludes the need for a separate validation set. BR networks generalize well (i.e. they tend not to be over-fitted) because they retain for training only the non-trivial links between nodes, forming a more parsimonious network than would be used in a fully connected back-propagation network [63]. Weight and bias values were updated according to Levenberg-Marquardt optimization (also known as 'damped least squares' [64,65]), an algorithm that also tends to produce results that generalize well. Performance during training was monitored via the sum of squared errors between the true and estimated classifications. Nodes in the hidden layer use a fast implementation equivalent of the hyperbolic tangent activation function, which has been shown to be superior to other sigmoid functions [66]. Nodes in the output layer of the 3-way classification use the softmax activation function; as this function exponentiates each input then divides by the sum of all exponentiated inputs, the outputs sum to unity and can be read as classification probabilities. Nodes in the output layer of the 2-way classification use the basic

sigmoid activation function, which 'squashes' output to a range between zero and one. This output can be read directly as the probability of an LSA classification (and, equivalently, as 1 minus the probability of an MSA classification).

For each network, classification probabilities (the probabilities that the assemblage is {LSA, MIS3 MSA, MIS4&5 MSA} or {LSA, MSA}) and an actual classification (corresponding to the highest of the probabilities) for each assemblage were recorded, as was the overall proportion of correct classifications. Whilst the percentage of correct classifications is the most relevant index of overall network performance, recording exact classification probabilities for individual assemblages allows for an assessment of the strength of each classification. Results for each assemblage can then be assessed in terms of 1) the distribution of exact probabilities, 2) the percentage of networks that correctly classify the assemblage, and 3) the overall classification based on the weighted mean probability over the whole ensemble of 1,000 networks. The weight attributed to a given network in this latter calculation is the proportion of correct classifications achieved by that network. The weighted mean classification is then the classification with the highest weighted mean probability.

## Typological indicators

As discussed above, the 'black box' nature of neural networks is often criticised. To extract as much information as possible from the trained networks, we modify a simulation method proposed by Baxt [59,60] that facilitates the extraction of delta values, which can be interpreted in a similar way to the effect sizes extracted from a multinomial logistic regression. A verbal description of this method is given here; more technical treatments are available in [59] and [60]. Once a given network has been trained, the original data can be fed through it to determine the resulting classification. To measure how differences in the presence or absence of a particular technology would alter the classification, the data for that technology at each of the 92 assemblages are inverted (i.e. where the true data indicates presence, this is inverted to absence, and vice versa). These partially inverted data are then fed through the trained network, and delta values are calculated by subtracting the result of the original classification from the result of the classification based on the partially inverted data. This process is repeated, inverting the data for only one technology at a time, for each of the 16 technologies, and for each of 1,000 trained networks. Results are then aggregated, and median delta values per technology are presented separately for changes from presence to absence and from absence to presence. 95% confidence intervals are constructed from the 2.5th and 97.5th percentiles of the sampling distribution of the median to give a measure of confidence in the median delta values for each technology in each of the two conditions. Due to the use of the softmax activation function in the output layers of the networks, median delta values can be interpreted as, for example, the median increase in the probability of an assemblage being categorised as LSA when a particular technology that was previously present (absent) is subtracted from (added to) that assemblage. Technologies for which the 95% confidence intervals of the median delta value do not encompass zero for a given period / industry are considered significant indicators (or contra-indicators) of that period / industry. To ensure that this approach is a rigorous as possible, a given technology is only considered a significant indicator if the two inversions (presence inverted to absence *and* absence inverted to presence) are of opposite sign and the 95% confidence intervals of neither encompass zero. Significance is assessed separately for each technology within each period / industry.

Sample sizes are monitored carefully to ensure that they are adequate for reliable results. Although there are 92 partially inverted assemblages entered into each network, the sample sizes for the two different inversions can vary considerably. For example, if a given technology

is present in only 5 assemblages, this will result in a sample size of 87 for the inversion from absence to presence, but a sample size of only 5 for the inversion from presence to absence. Even when passed through 1,000 different networks, a sample size this small is insufficient to produce a reliable result. Following the logic outlined by Baxt and White [60], a sample size of ≥10 is regarded as being sufficient to produce a reliable result. Final conclusions regarding the values of technologies as indicators of a given period / industry therefore take into account both the significance of the (paired) delta values and the sample sizes from which they were derived. All analyses were carried out in Matlab R2018a (Mathworks, Natick, MA., USA); all code is available as S1 and S2 Codes associated with this paper.

## Results

### Assemblage classification

**3-way classification.** The overall performance of the weighted mean ensemble of 1,000 networks for the 3-way classification is shown in Table 1; the ensemble misclassified only 5 assemblages, leading to an overall accuracy of 94.57%. A summary of the performance of each individual network in classifying each of the 92 assemblages is shown in Fig 4. The figure shows the distributions of exact classification probabilities into each of the three industries / periods for each of the 92 assemblages produced by each of the 1,000 networks. A correct classification occurs when the probability of correct classification is greater than the probability of either incorrect classification. For example, if an LSA assemblage has probabilities $p_{LSA}$ = .5, $p_{MIS3\&4MSA}$ = .25, and $p_{MIS5MSA}$ = .25, it is correctly classified. As there are three possible classifications, any LSA assemblage with $p_{LSA} > \frac{1}{3}$ *could* be classified correctly (provided both $p_{MIS3\&4MSA}$ and $p_{MIS5MSA}$ are $<p_{LSA}$), and any LSA assemblage with $p_{LSA} > \frac{1}{2}$ is *necessarily* classified correctly. Fig 5 shows the exact classification probabilities calculated via the weighted mean of the ensemble of 1,000 networks, with incorrectly classified assemblages labelled. Percentages of correct classifications across the 1,000 networks for each assemblage are shown in Fig 6; in this figure, assemblages incorrectly classified by the weighted median ensemble of networks are shown in red. Details of the incorrectly classified sites are shown in Table 2.

**2-way classification.** The overall performance of the weighted mean ensemble of 1,000 networks for the 2-way classification is shown in Table 3; the ensemble misclassified only 1 assemblage, leading to an overall accuracy of 98.91%. A summary of the performance of each individual network in classifying each of the 92 assemblages is shown in Fig 7. The figure shows the distributions of exact classification probabilities into each of the three industries / periods for each of the 92 assemblages produced by each of the 1,000 networks. Percentages of correct classifications across the 1,000 networks for each assemblage are shown in Fig 8; in this

**Table 1. Confusion table (3-way).**

| | | Target Class | | | |
|---|---|---|---|---|---|
| | | LSA | MIS3&4 MSA | MIS5 MSA | Specificity (TN%) |
| Output Class | LSA | 30 | 0 | 0 | 100.00 |
| | MIS3&4 MSA | 1 | 30 | 3 | 88.24 |
| | MIS5 MSA | 0 | 1 | 27 | 96.43 |
| | Sensitivity (TP%) | 96.77 | 96.77 | 90.00 | |
| | | | | Accuracy (%) | 94.57 |

Confusion table for the overall classifications produced by the ensemble of 1,000 networks in the 3-way analysis. TP% = percentage of true positives. TN% = percentage of true negatives.

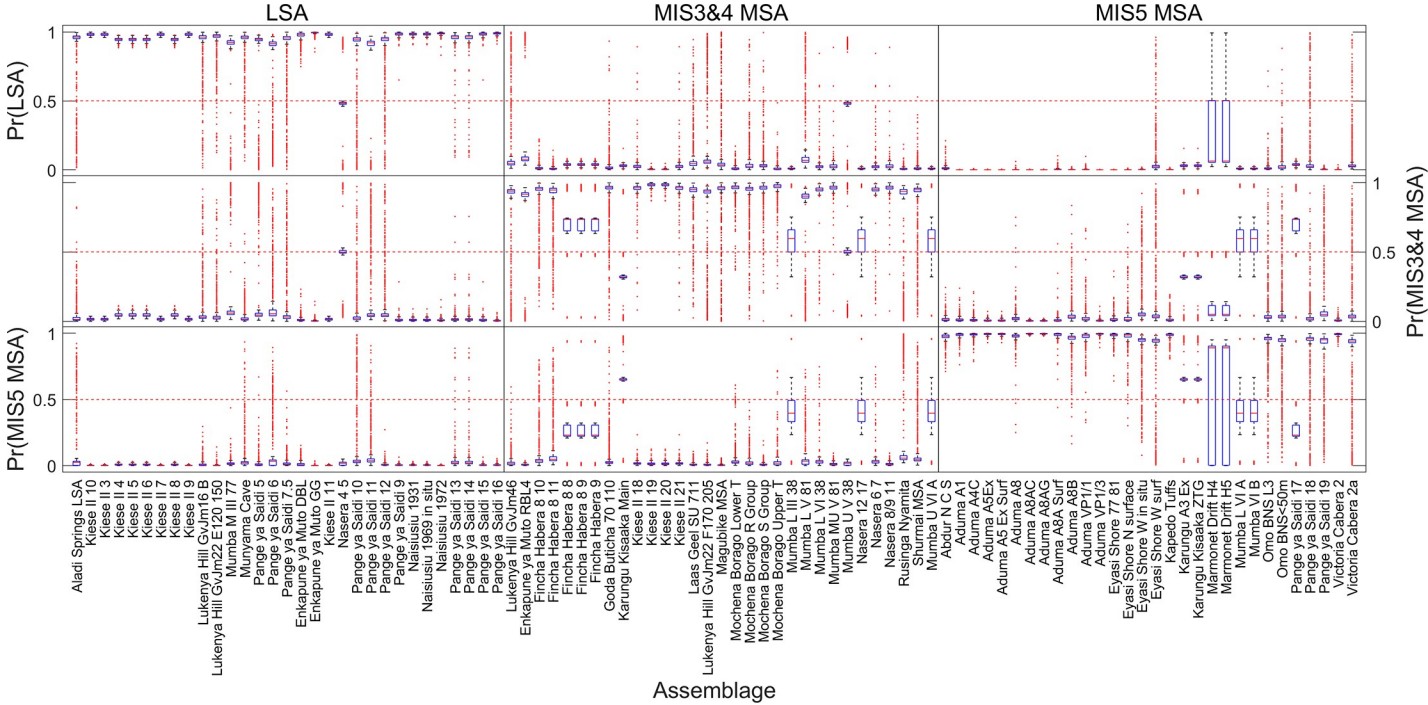

**Fig 4. Boxplots of classification probabilities (3-way).** Classification probabilities are derived from each of 1,000 neural networks in the 3-way analysis. Red lines show medians, boxes extend to the inter-quartile range, and whiskers show 95% of the distribution. Outliers are shown as red dots. For correct classification, values on the diagonal should be as close to 1 as possible (e.g. values of Pr (LSA) in the LSA section should be as close to 1 as possible, etc.).

figure, assemblages incorrectly classified by the weighted median ensemble of networks are shown in red. There was only one incorrectly classified assemblage in the 2-way analysis, Nasera 4/5, an LSA assemblage misclassified as MIS3&4 MSA.

## Typological indicators

**3-way classification.** Distributions of 1,000 bootstrap estimates of delta values for both addition and removal and for each of the 16 technologies in the 3-way analyses are presented in Fig 9; median values and 95% confidence intervals are provided for LSA indicators in Table 4, MIS3&4 MSA indicators in Table 5, and MIS5 MSA indicators in Table 6. Overall, 6 technologies were significant indicators or contra-indicators of at least one class; these are summarised in Table 7. These results can be summarised by noting that:

- LSA assemblages are indicated by the presence of Backed Pieces, bipolar reduction, and blades, and by the absence of core tools, Levallois flakes and point technology;

- MIS3&4 MSA assemblages are indicated by the presence of point technology, and by the absence of core tools;

- MIS5 MSA assemblages are indicated by the presence of core tools, and by the absence of Backed Pieces.

**2-way classification.** Distributions of 1,000 bootstrap estimates of delta values for both addition and removal and for each of the 16 technologies in the 2-way analyses are presented in Fig 10; median values and 95% confidence intervals are provided for LSA indicators in

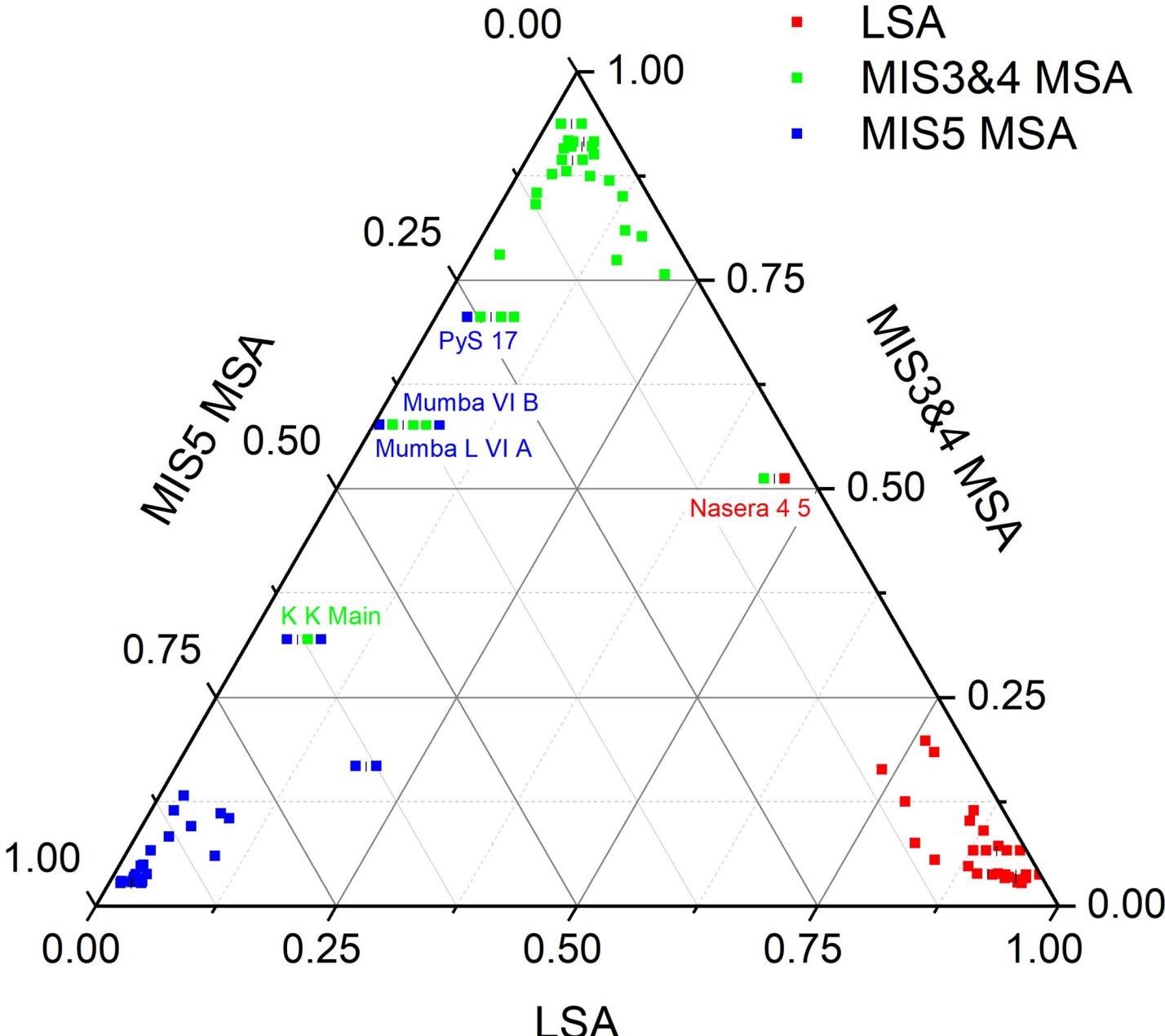

**Fig 5. Ternary plot of exact classification probabilities (3-way).** Exact classification probabilities for the 92 assemblages in the 3-way analysis via the weighted mean of the ensemble of 1,000 networks. The location of a point shows simultaneously the probabilities of it being classified as LSA, MIS3&4 MSA, and MIS5 MSA. Correctly classified assemblages will be at or close to the maximum for their respective axis. Incorrectly classified assemblages are labelled. Where multiple assemblages plot in the same location they are re-arranged horizontally around the bar (|) symbol that shows their true location.

Table 8 (as this is a 2-way analysis, the equivalent MSA results are the inverse of Table 8, and are not shown). Overall, 7 technologies were significant indicators or contra-indicators of either the LSA or MSA, as summarised in the final column of Table 7. These results can be summarised by noting that:

• LSA assemblages are indicated by the presence of Backed Pieces, bipolar technology, and blades;

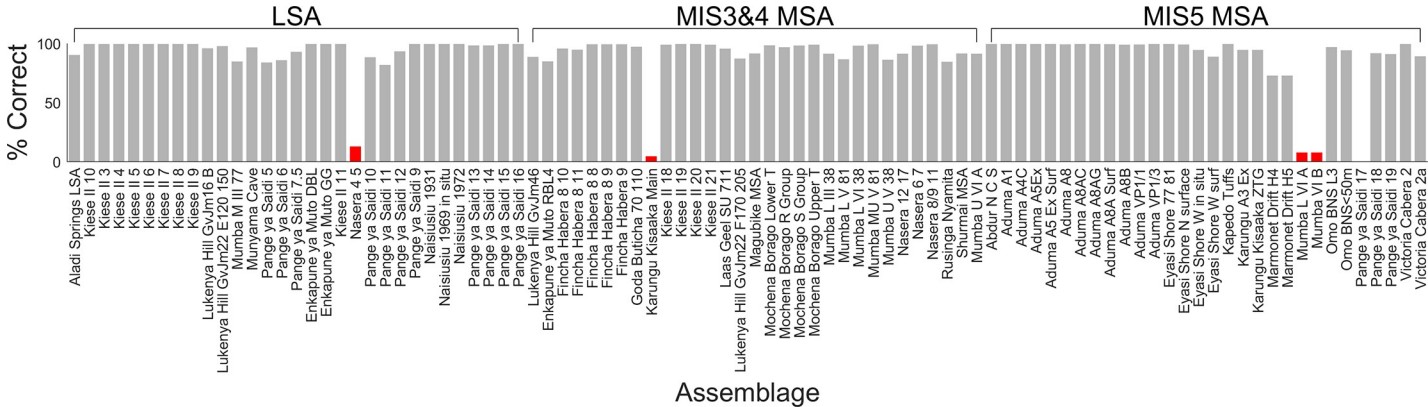

**Fig 6. Percentages of correct classification (3-way).** Percentages of the 1,000 networks that classify each assemblage correctly in the 3-way analysis. Red bars indicate those assemblages classified incorrectly by a weighted mean over the ensemble of networks. Note the distinction between the percentage of individual networks that classify an assemblage correctly and the overall classification of an assemblage by the weighted mean of the whole ensemble of networks.

- MSA assemblages are indicated by the presence of core tools, Levallois flakes, point technology, and scrapers.

## Discussion

The following sections provide more detail on the technologies found to be significant predictors of one or more classes in the analyses above, and on possible reasons for the incorrect classifications of assemblages where they occur. The discussion then moves on to address broader issues of typology and quantification in relation to the LSA / MSA transition in eastern Africa.

### Significant lithic components

**Backed pieces.** Backed Pieces appear in a high proportions in both LSA (77%) and MIS3&4 MSA (71%) assemblages, and are considerably less prevalent in MIS5 MSA assemblages (23%). The 3-way analysis establishes them as indicators of LSA and as contra-indicators of MIS5 MSA. In the 2-way analyses, backed pieces are established as an indicator of the LSA. However, the presence of backed pieces in 71% of the MIS3&4 MSA assemblages in the dataset demonstrates that backing is certainly not an exclusively LSA phenomenon. In this context, Leplongeon [67] highlights an important distinction between the use of backing as a form of retouch and its place within systematic production of microlithic tools. In her study of the Porc Epic and Goda Buticha assemblages of Ethiopia she suggests that a clear intention to systematically produce microliths remains a hallmark of the LSA; she also notes, however, that microliths are not as numerous in eastern African LSA assemblages as they are in LSA assemblages from elsewhere in Africa.

**Table 2. Assemblages misclassified in the 3-way analysis.**

| | Classification | |
| --- | --- | --- |
| Assemblage | Target | Estimate |
| Nasera 4 5 | LSA | MIS3&4 MSA |
| Karungu Kisaaka Main | MIS3&4 MSA | MIS5 MSA |
| Mumba L VI A | MIS5 MSA | MIS3&4 MSA |
| Mumba VI B | MIS5 MSA | MIS3&4 MSA |
| Panga ya Saidi 17 | MIS5 MSA | MIS3&4 MSA |

**Table 3. Confusion table (2-way).**

| | | Target Class | | |
|---|---|---|---|---|
| | | LSA | MSA | Specificity (TN%) |
| Output Class | LSA | 30 | 0 | 100.00 |
| | MSA | 1 | 61 | 98.39 |
| | Sensitivity (TP%) | 96.77 | 100.00 | |
| | | | Accuracy (%) | 98.91 |

Confusion table for the overall classifications produced by the ensemble of 1,000 networks in the 2-way analysis. TP% = percentage of true positives. TN% = percentage of true negatives.

**Bipolar technology.** Only 30% of the MIS5 MSA assemblages in this dataset contain examples of bipolar technology; this figure rises to 74% for MIS3&4 MSA and 81% for LSA assemblages. Although the 3-way analysis extracts bipolar technology as an indicator of LSA assemblages it should be noted that such technology is also highly prevalent in MIS3&4 MSA assemblages, and that the 3-way analysis does not explicitly identify bipolar technology as contra-indicator for either of the other classes. The 2-way analysis extracts bipolar technology as an indicator of the LSA rather than the MSA taken as a whole, but this analysis is of course insensitive to changes in assemblage composition that take place *within* the MSA. Overall, 52% percent of all MSA assemblages surveyed contain evidence of bipolar reduction, a substantially smaller percentage than found in the LSA (81%).

**Blade technology.** Blade technology is abundant in this dataset, being present in 84% of all assemblages; percentages vary from 73% in MIS5 MSA to 84% in MIS3&4 MSA and 94% in the LSA. The ANN analyses consistently reveal blade technology as an indicator of the LSA. In this case, as with the majority of cases outlined here, the indicator must be viewed as a threshold frequency rather than a simple binary indicator; MSA assemblages are not indicated by the *absence* of blades, but fewer contain blades than LSA assemblages.

**Core tools.** Core tools are present in 25% of assemblages overall, declining markedly from MIS5 MSA (57%) through MIS3&4 MSA (16%) to LSA (3%). The 2-way analysis establishes core tools as indicators of the MSA as opposed to the LSA, but the finer-grained 3-way analysis

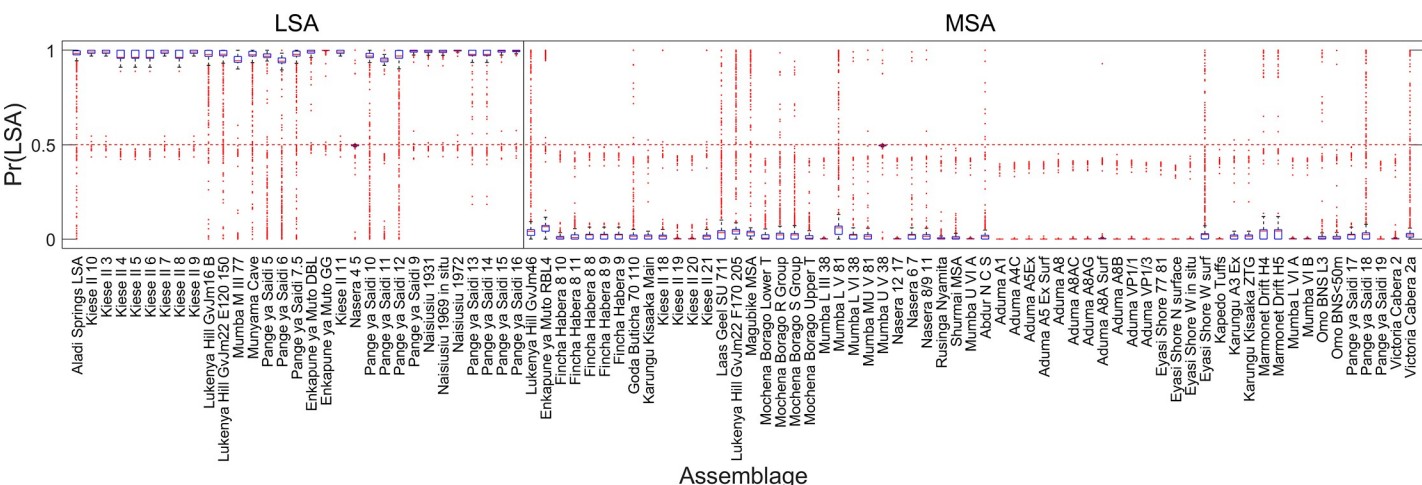

**Fig 7. Boxplots of classification probabilities (2-way).** Classification probabilities are derived from each of 1,000 neural networks in the 2-way analysis. Red lines show medians, boxes extend to the inter-quartile range, and whiskers show 95% of the distribution. Outliers are shown as red dots. For correct classification, values for the LSA assemblages should be close to 1 and those for the MSA assemblages close to zero.

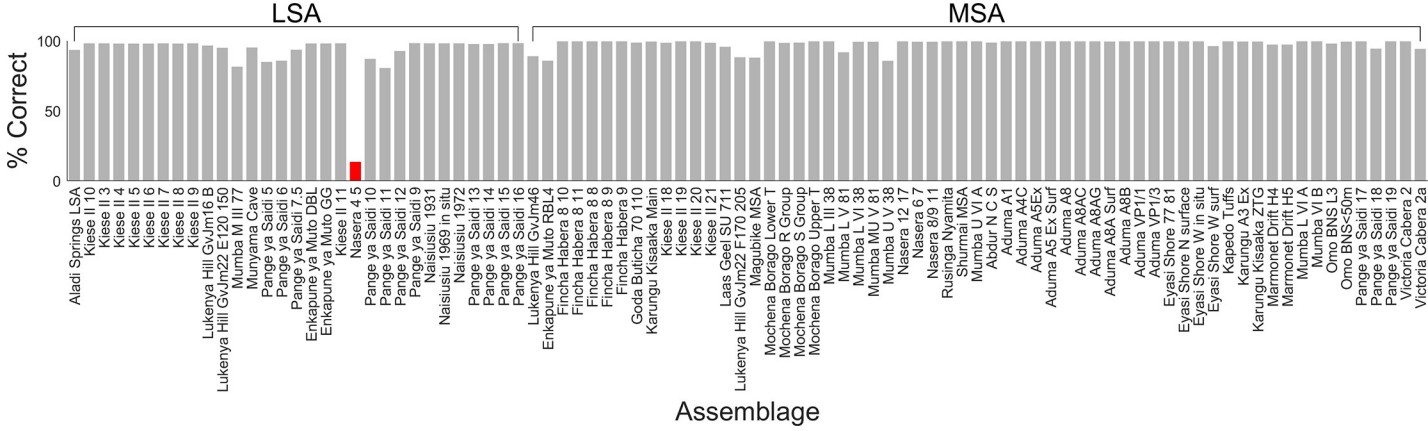

**Fig 8. Percentages of correct classification (2-way).** Percentages of the 1,000 networks that classify each assemblage correctly in the 2-way analysis. Red bars indicate those assemblages classified incorrectly by a weighted mean over the ensemble of networks.

establishes them as indicators of MIS5 MSA and as contra-indicators of both MIS3&4 MSA and LSA assemblages. While both results are consistent with the above given percentages, the latter provides a clearer picture of their greater prevalence in MIS5.

**Levallois flakes.** Levallois flake technology is present in all assemblage classes at relatively high frequencies: 26% in LSA, 71% in MIS3&4 MSA, and 83% in MIS4&5 MSA assemblages, with an overall percentage in the whole dataset of 60%. The 3-way ANN analyses extract it as a

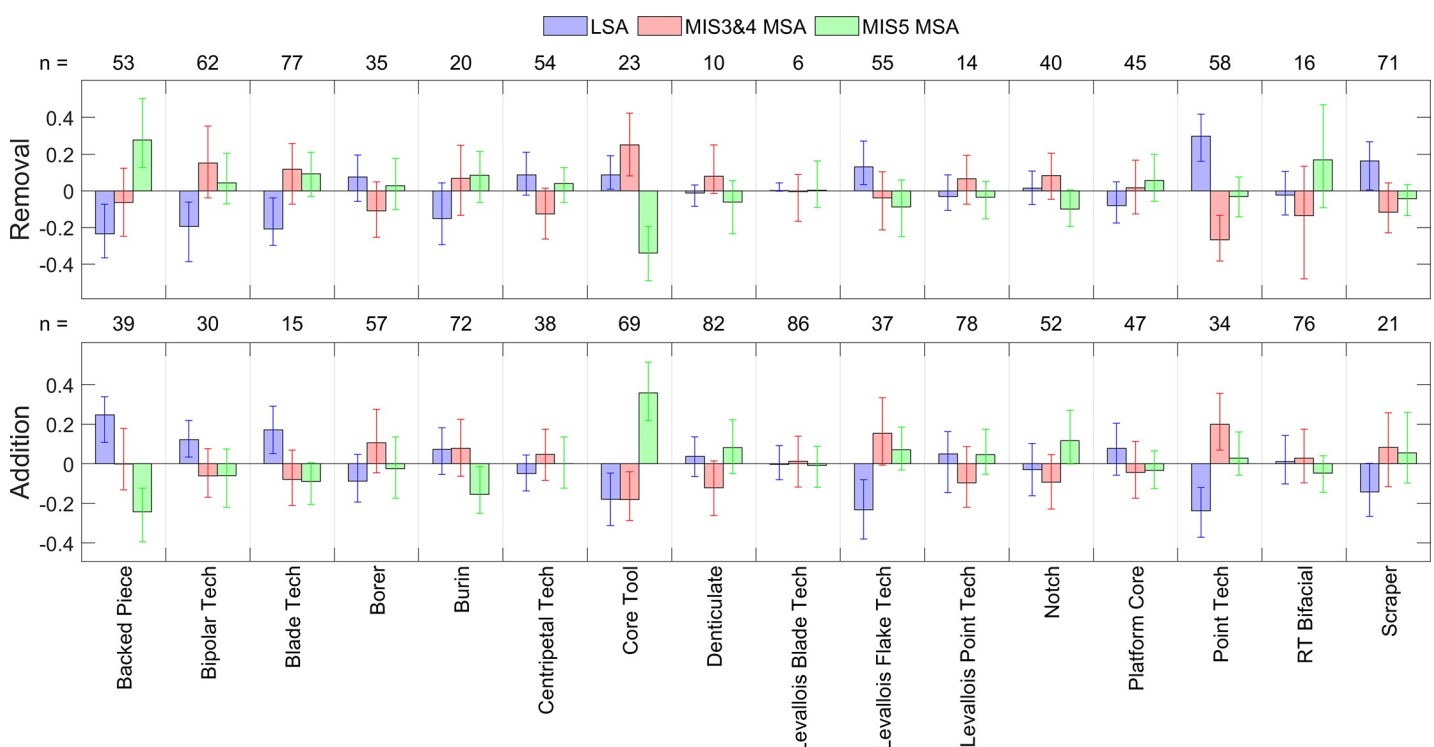

**Fig 9. Delta values (3-way).** Plots of delta values for the 16 technologies via the bootstrap procedure in the 3-way analysis. Positive values in the 'Removal' graph indicate that removing a given technology increases the probability of classification to a given period / industry. Positive values in the 'Addition' graph indicate that adding a given technology increases the probability of classification to a given period / industry. Error bars are 2.5 and 97.5 percentiles.

**Table 4. Significant indicators of LSA assemblages in the 3-way analysis.**

| Lithic component | False Absence | | | | False Presence | | | | Sig | Ind / Contra |
|---|---|---|---|---|---|---|---|---|---|---|
| | Lower | Median | Upper | n | Lower | Median | Upper | n | | |
| Backed Pieces | -0.365 | -0.236 | -0.074 | 53 | 0.107 | 0.247 | 0.339 | 39 | SIG | IND |
| Bipolar Tech | -0.386 | -0.195 | -0.062 | 62 | 0.035 | 0.122 | 0.218 | 30 | SIG | IND |
| Blade Tech | -0.297 | -0.208 | -0.038 | 77 | 0.051 | 0.172 | 0.291 | 15 | SIG | IND |
| Borer | -0.058 | 0.076 | 0.195 | 35 | -0.193 | -0.087 | 0.049 | 57 | NS | |
| Burin | -0.294 | -0.150 | 0.043 | 20 | -0.055 | 0.073 | 0.182 | 72 | NS | |
| Centripetal Tech | -0.024 | 0.086 | 0.210 | 54 | -0.137 | -0.049 | 0.045 | 38 | NS | |
| Core Tool | 0.009 | 0.086 | 0.191 | 23 | -0.311 | -0.180 | -0.048 | 69 | SIG | CONTRA |
| Denticulate | -0.085 | -0.013 | 0.032 | 10 | -0.065 | 0.037 | 0.136 | 82 | NS | |
| Levallois Blade Tech | -0.003 | 0.001 | 0.043 | 6 | -0.081 | -0.003 | 0.091 | 86 | NS | |
| Levallois Flake Tech | 0.034 | 0.131 | 0.271 | 55 | -0.380 | -0.231 | -0.081 | 37 | SIG | CONTRA |
| Levallois Point Tech | -0.107 | -0.031 | 0.086 | 14 | -0.146 | 0.049 | 0.163 | 78 | NS | |
| Notch | -0.076 | 0.015 | 0.108 | 40 | -0.161 | -0.029 | 0.102 | 52 | NS | |
| Platform Core | -0.176 | -0.080 | 0.049 | 45 | -0.058 | 0.078 | 0.205 | 47 | NS | |
| Point Tech | 0.161 | 0.297 | 0.418 | 58 | -0.372 | -0.237 | -0.120 | 34 | SIG | CONTRA |
| RT Bifacial | -0.132 | -0.023 | 0.106 | 16 | -0.101 | 0.010 | 0.144 | 76 | NS | |
| Scraper | 0.005 | 0.163 | 0.267 | 71 | -0.266 | -0.143 | 0.002 | 21 | NS | |

IND = significant indicator; CONTRA = significant contra-indicator. Technologies with no value in the 'Indicator' column fail to reach significance due to small sample size and / or the fact that their 95% confidence intervals include zero.

contra-indicator of LSA assemblages, whilst the 2-way analyses establish it as indicating MSA rather than LSA assemblages. Both inferences are robust, but it must be noted that Levallois flakes remain very much a part of LSA assemblages, albeit appearing in far fewer assemblages of this industry.

**Table 5. Significant indicators of MIS3&4 MSA assemblages in the 3-way analysis.** Details as per Table 4.

| Lithic component | False Absence | | | | False Presence | | | | Sig | Ind / Contra |
|---|---|---|---|---|---|---|---|---|---|---|
| | Lower | Median | Upper | n | Lower | Median | Upper | n | | |
| Backed Pieces | -0.248 | -0.064 | 0.123 | 53 | -0.131 | -0.001 | 0.179 | 39 | NS | |
| Bipolar Tech | -0.039 | 0.152 | 0.354 | 62 | -0.168 | -0.062 | 0.075 | 30 | NS | |
| Blade Tech | -0.072 | 0.117 | 0.258 | 77 | -0.211 | -0.078 | 0.069 | 15 | NS | |
| Borer | -0.253 | -0.109 | 0.050 | 35 | -0.045 | 0.105 | 0.275 | 57 | NS | |
| Burin | -0.134 | 0.068 | 0.249 | 20 | -0.063 | 0.078 | 0.223 | 72 | NS | |
| Centripetal Tech | -0.264 | -0.127 | 0.015 | 54 | -0.084 | 0.048 | 0.174 | 38 | NS | |
| Core Tool | 0.081 | 0.250 | 0.423 | 23 | -0.287 | -0.181 | -0.040 | 69 | SIG | CONTRA |
| Denticulate | -0.014 | 0.079 | 0.251 | 10 | -0.262 | -0.120 | 0.015 | 82 | NS | |
| Levallois Blade Tech | -0.166 | -0.004 | 0.089 | 6 | -0.117 | 0.013 | 0.140 | 86 | NS | |
| Levallois Flake Tech | -0.214 | -0.038 | 0.105 | 55 | -0.006 | 0.154 | 0.332 | 37 | NS | |
| Levallois Point Tech | -0.073 | 0.065 | 0.193 | 14 | -0.219 | -0.096 | 0.087 | 78 | NS | |
| Notch | -0.047 | 0.083 | 0.206 | 40 | -0.229 | -0.093 | 0.047 | 52 | NS | |
| Platform Core | -0.127 | 0.017 | 0.167 | 45 | -0.173 | -0.043 | 0.112 | 47 | NS | |
| Point Tech | -0.384 | -0.267 | -0.134 | 58 | 0.069 | 0.199 | 0.356 | 34 | SIG | IND |
| RT Bifacial | -0.481 | -0.135 | 0.134 | 16 | -0.097 | 0.029 | 0.175 | 76 | NS | |
| Scraper | -0.230 | -0.117 | 0.044 | 71 | -0.115 | 0.083 | 0.258 | 21 | NS | |

**Table 6. Significant indicators of MIS5 MSA assemblages in the 3-way analysis.** Details as per Table 4.

| Lithic component | False Absence | | | | False Presence | | | | Sig | Ind / Contra |
|---|---|---|---|---|---|---|---|---|---|---|
| | Lower | Median | Upper | n | Lower | Median | Upper | n | | |
| Backed Pieces | 0.126 | 0.277 | 0.503 | 53 | -0.393 | -0.243 | -0.122 | 39 | SIG | CONTRA |
| Bipolar Tech | -0.070 | 0.043 | 0.204 | 62 | -0.220 | -0.059 | 0.074 | 30 | NS | |
| Blade Tech | -0.032 | 0.093 | 0.210 | 77 | -0.206 | -0.090 | 0.005 | 15 | NS | |
| Borer | -0.101 | 0.027 | 0.177 | 35 | -0.174 | -0.024 | 0.136 | 57 | NS | |
| Burin | -0.063 | 0.084 | 0.214 | 20 | -0.250 | -0.154 | -0.013 | 72 | NS | |
| Centripetal Tech | -0.063 | 0.039 | 0.127 | 54 | -0.122 | 0.000 | 0.136 | 38 | NS | |
| Core Tool | -0.490 | -0.340 | -0.195 | 23 | 0.217 | 0.358 | 0.512 | 69 | SIG | IND |
| Denticulate | -0.235 | -0.062 | 0.057 | 10 | -0.049 | 0.082 | 0.222 | 82 | NS | |
| Levallois Blade Tech | -0.089 | 0.002 | 0.162 | 6 | -0.119 | -0.008 | 0.089 | 86 | NS | |
| Levallois Flake Tech | -0.249 | -0.087 | 0.058 | 55 | -0.032 | 0.070 | 0.186 | 37 | NS | |
| Levallois Point Tech | -0.154 | -0.035 | 0.051 | 14 | -0.053 | 0.047 | 0.175 | 78 | NS | |
| Notch | -0.194 | -0.099 | 0.005 | 40 | -0.002 | 0.117 | 0.270 | 52 | NS | |
| Platform Core | -0.057 | 0.057 | 0.199 | 45 | -0.124 | -0.033 | 0.066 | 47 | NS | |
| Point Tech | -0.141 | -0.032 | 0.075 | 58 | -0.057 | 0.029 | 0.161 | 34 | NS | |
| RT Bifacial | -0.093 | 0.169 | 0.469 | 16 | -0.143 | -0.047 | 0.041 | 76 | NS | |
| Scraper | -0.136 | -0.043 | 0.034 | 71 | -0.097 | 0.054 | 0.260 | 21 | NS | |

**Point technology.** Point technology is present in 63% of assemblages; it occurs in 32% of LSA assemblages, 84% of MIS3&4 MSA assemblages and 73% of MIS5 MSA assemblages. It is established in the 3-way analysis as an indicator of MIS3&4 MSA and as a contra-indicator of LSA; the 2-way analysis establishes it as an indicator of the MSA.

**Scrapers.** Scrapers are present in 77% of all assemblages in the dataset; 77% of MIS5 MSA, 81% of MIS3&4 MSA, and 74% of LSA assemblages contain this technology. Although scrapers do not appear as a significant indicator in the 3-way analysis, the 2-way analysis establishes them as an indicator of the MSA due to their overall higher percentage in all MSA (79%) than in LSA (74%) assemblages.

## Incorrectly classified assemblages

The 3-way analyses incorrectly classified five assemblages: Nasera 4/5 (LSA, misclassified as MIS3&4 MSA), Karungu Kisaaka Main (MIS3&4 MSA, incorrectly classified as MIS5 MSA), Mumba L VI A, Mumba VI B, and Panga ya Saidi 17 (all MIS5 MSA, misclassified as MIS3&4 MSA). The 2-way analyses incorrectly classified only Nasera 4/5, again as MSA rather than LSA. These incorrect classifications are marginal, with relatively high classification probabilities for more than one class (see Figs 5 and 6). In the 3-way analyses Nasera 4/5 has an

**Table 7. Summary of significant indicators and contra-indicators from the 3-way (first three columns) and 2-way (final column) analyses.** Details as per Table 4.

| Lithic component | 2-Way | 3-Way | | |
|---|---|---|---|---|
| | | LSA | MIS3&4 MSA | MIS5 MSA |
| Backed Pieces | LSA | IND | | CONTRA |
| Bipolar Tech | LSA | IND | | |
| Blade Tech | LSA | IND | | |
| Core Tool | MSA | CONTRA | CONTRA | IND |
| Levallois Flake Tech | MSA | CONTRA | | |
| Point Tech | MSA | CONTRA | IND | |
| Scraper | MSA | | | |

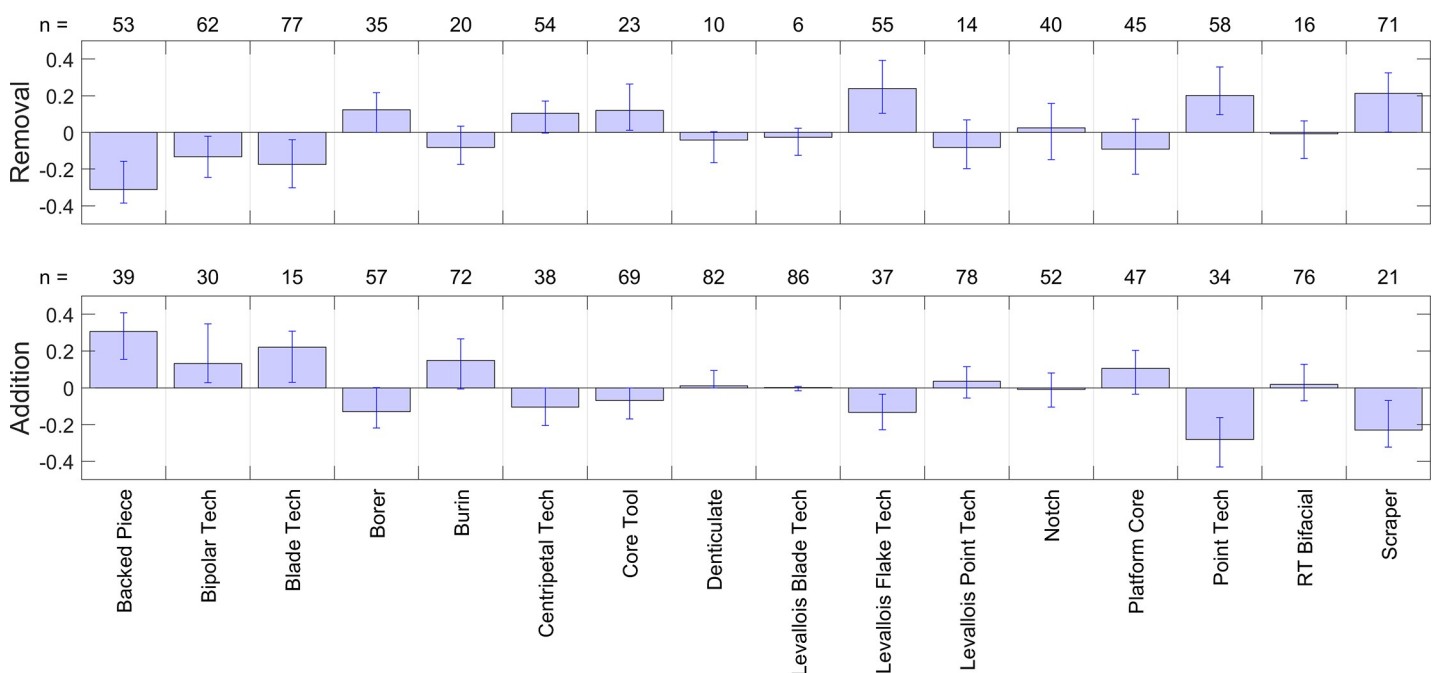

**Fig 10. Delta values (2-way).** Plots of delta values for the 16 technologies via the bootstrap procedure in the 2-way analysis. Positive values in the 'Removal' graph indicate that removing a given technology increases the probability of MSA classification. Positive values in the 'Addition' graph indicate that adding a given technology increases the probability of LSA classification. Error bars are 2.5 and 97.5 percentiles.

MIS3&4 MSA classification probability of .526, with a .458 probability of LSA classification. Mumba L VI A and Mumba VI B, which are identical in terms of composition, have MIS3&4 MSA probabilities of .595, and MIS5 probabilities of .396. Karungu Kisaaka Main has a

**Table 8. Significant indicators of LSA and MSA assemblages in the 2-way analysis.**

| Lithic component | False Absence | | | | False Presence | | | | Sig | Ind / Contra |
|---|---|---|---|---|---|---|---|---|---|---|
| | Lower | Median | Upper | n | Lower | Median | Upper | n | | |
| Backed Pieces | -0.384 | -0.311 | -0.157 | 53 | 0.154 | 0.305 | 0.408 | 39 | SIG | IND |
| Bipolar Tech | -0.245 | -0.132 | -0.020 | 62 | 0.027 | 0.132 | 0.348 | 30 | SIG | IND |
| Blade Tech | -0.303 | -0.174 | -0.040 | 77 | 0.030 | 0.220 | 0.307 | 15 | SIG | IND |
| Borer | 0.000 | 0.122 | 0.216 | 35 | -0.219 | -0.129 | 0.000 | 57 | NS | |
| Burin | -0.174 | -0.082 | 0.035 | 20 | -0.006 | 0.149 | 0.267 | 72 | NS | |
| Centripetal Tech | -0.005 | 0.105 | 0.172 | 54 | -0.205 | -0.105 | -0.001 | 38 | NS | |
| Core Tool | 0.012 | 0.119 | 0.264 | 23 | -0.170 | -0.069 | -0.001 | 69 | SIG | CONTRA |
| Denticulate | -0.164 | -0.042 | 0.004 | 10 | -0.001 | 0.010 | 0.095 | 82 | NS | |
| Levallois Blade Tech | -0.125 | -0.026 | 0.022 | 6 | -0.016 | 0.002 | 0.007 | 86 | NS | |
| Levallois Flake Tech | 0.105 | 0.239 | 0.392 | 55 | -0.227 | -0.134 | -0.036 | 37 | SIG | CONTRA |
| Levallois Point Tech | -0.198 | -0.082 | 0.069 | 14 | -0.056 | 0.036 | 0.114 | 78 | NS | |
| Notch | -0.149 | 0.024 | 0.158 | 40 | -0.105 | -0.009 | 0.081 | 52 | NS | |
| Platform Core | -0.229 | -0.090 | 0.073 | 45 | -0.034 | 0.105 | 0.205 | 47 | NS | |
| Point Tech | 0.096 | 0.201 | 0.356 | 58 | -0.430 | -0.281 | -0.162 | 34 | SIG | CONTRA |
| RT Bifacial | -0.143 | -0.007 | 0.063 | 16 | -0.070 | 0.019 | 0.128 | 76 | NS | |
| Scraper | 0.002 | 0.212 | 0.324 | 71 | -0.323 | -0.229 | -0.069 | 21 | SIG | CONTRA |

LSA = significant indicator; MSA = significant contra-indicator. Further details as per Table 4.

MIS3&4 MSA probability of .318, and an MIS5 MSA probability of .652. Panga ya Saidi has a MIS3&4 probability of .734 and a MIS5 probability of .229. In the 2-way analysis, Nasera 4 5 has an LSA probability of .497 and an MSA probability of .503. These sites are misclassified because they contain complex combinations of technologies that are established as indicative of multiple classes. In this context Nasera 4/5, a particularly rich assemblage in terms of technological diversity, acts as an illustrative example.

The Nasera 4/5 assemblage contains all three indicators of the LSA established via the 2-way analysis (backed pieces, bipolar technology, and blades); however, it also contains three technologies identified an indicators of the MSA via this analysis (Levallois flakes, point technology, and scrapers). This assemblage therefore contains a variety of contradictory signals which lead to its marginal misclassification in both the 2- and 3-way analyses. A more pressing issue is that, given the presence-absence typology used, the Nasera 4/5 assemblage is identical to that from Mumba U V 38, with the latter classified as MIS3&4 MSA. Neural networks are 'universal approximators' in that they can successfully solve any complex classification topology given a sufficient number of nodes and sufficiently large and non-overlapping classification sets. This example shows that the classification sets in this dataset in fact overlap, and of course it is impossible for a neural network (or any other classification algorithm) to assign two identical assemblages to different classes. This issue also occurs in the following cases: Mumba L VI A and Mumba VI B (both MIS5 MSA) are identical to Mumba U VI A, Mumba L III 38, and Nasera 12 17 (all MIS3&4 MSA); Karungu Kisaaka Main (MIS3&4 MSA) is identical to Karungu A3 Ex and Karungu Kisaaka ZTG (both MIS5 MSA); Panga ya Saidi 17 (MIS5 MSA) is identical to Fincha Habrera 8 8, 8 9, and 9 (all MIS3&4 MSA). These instances account for all five cases of misclassification by the networks detailed above.

In some cases there is recent archaeological or chronological evidence suggesting that the purported 'misclassification' of a given assemblage may actually provide a more realistic designation. As regards Nasera 4/5, for example, recent radiocarbon dating of ostrich egg shell (OES) fragments from the deposits above and below this layer by Ranhorn and Tryon [5], coupled with earlier amino acid racemization dates from OES fragments within layer 4, suggest that this assemblage may be rather older than originally thought. Whilst chronological age cannot be taken to directly indicate an industrial affiliation, the misclassification of Nasera 4/5 as MSA rather than LSA here is consistent with Ranhorn and Tryon's [5] contention that this level is older than originally described by Mehlman [68], and perhaps close to the older end of the 32.5 – 44ka bracket derived by Kokis [69].

With reference to the Karungu Kisaaka Main assemblage, both Blegen and colleagues [70] and Tryon [12] suggest that assemblages within the Lake Victoria basin demonstrate the persistence of what are here regarded as MIS5 MSA technologies well into later Marine Isotope Stages. Tryon [12] suggests that the high level of endemism apparent among faunal species in the Lake Victoria basin reflects the relative isolation of this region, perhaps due to environmental factors. If the human inhabitants of the basin were similarly isolated, this could account for the observed delay in technological developments [12]. The notion of delayed technological development in this region is supported by the 'misclassification' of the Karunga Kisaaka Main assemblage as MIS5 MSA; based on the typology adopted here, this assemblage is identical to the (MIS5) Karungu ZTG and A3 Ex assemblages, consistent with the notion of nominally MIS5 MSA technological components persisting into MIS4 and beyond.

The above issues may be seen as problems associated with quantification and typological analysis, but in fact they serve as indicators of a much wider issue in that many assemblages do indeed show indicators of more than one period. Quantifying indicators in this way makes explicit the problems of trying to discretize what is, in fact, continuous variation between the MSA and LSA, with many 'LSA' elements appearing in the MSA, and many 'MSA' elements

persisting well into the LSA. By way of further illustration of this issue, it is interesting to note that in the 3-way analyses, 43% of assemblages contain indicators of more than one class, with 18% of assemblages containing indicators of all three classes. 45% of assemblages contain contra-indicators of more than one class, with 12% containing contra-indicators of all three classes. In the 2-way analyses, 88% of assemblages contain indicators of both LSA and MSA.

## Typology, quantification, and the LSA / MSA transition

Two of the most important results of the foregoing analyses are, in a sense, negative. Firstly, of the 16 technologies included in the analyses, only 7 were effective in distinguishing between classes of assemblage. In both sets of analyses, therefore, the majority of technologies did not differ significantly between classes. Secondly, no single technology acts alone as a *fossil directeur* of any particular class; instead, constellations of co-occurring technologies increase the probability that the assemblage of which they are components will be classified in a particular way. Taken together, these results suggest that there is more continuity than change in the lithic assemblages of MIS2-5 in eastern Africa. Extracting those technologies that *do* distinguish between classes, however, leads to models that misclassify less than 5% of assemblages, demonstrating that it is possible to establish signatures of change both within the MSA and between the MSA and the LSA. The ability of neural networks to pick out significant predictors from a background of relative continuity suggests that they have clear advantages over traditional parametric statistics that rely on normally distributed data and linear relationships between variables, and that often fail to model nonlinear interactions between variables. The success of the classifications reported above suggests that neural networks should be embraced as a valuable set of tools for the analysis of archaeological data.

The transition from the MSA to the LSA is a protracted process, with one recent estimate suggesting that it occurs over a minimum of 5-10 ka [12]. The results presented above suggest that this estimate should certainly be viewed as a minimum, though it remains possible that the transition could have occured rapidly within individual, smaller areas within our study region. They further suggest that the transition remains difficult to define, but that constellations of artefacts that indicate particular periods or industries can be used to situate assemblages reliably along the transition. Although the analyses reported above are capable of doing this for the assemblages *within the current dataset*, such methods should not be seen as substitute for direct geochronological determinations; further research into the methods employed here may lead to the development of analytical tools that facilitate the classification of assemblages into industrial complexes (i.e. LSA, MSA), but at present these results should not be viewed as providing a predictive tool.

It would, perhaps, be ideal to ignore the labels attributed to particular assemblages at the outset and to rely instead on their chronological placement, although this is partially constrained by the resolution of dating available. However, we may legitimately ask whether a strictly chronological approach to the transition would offer greater explanatory power than a typological one, and the extent to which it may obscure patterns of intra-regional technological persistence, such as highlighted in the Lake Victoria basin above. An interesting parallel is provided by palaeontological taxonomy, particularly in relation to anagenesis and cladogenesis. If the transition from MSA to LSA can be regarded as a cultural analogue of anagenesis – that is, if the MSA gradually morphs into the LSA over time – then the distinction between the two is as arbitrary as that between, for example, European *Homo heidelbergensis* and the Neanderthals. Early *H. heidelbergensis* and late Neanderthals are relatively easily recognised, but the distinctions between late *H. heidelbergensis* and early Neanderthals are much harder to draw. With anagenesis, taxonomy involves imposing an arbitrary division between species that in

fact show continuous variation through time. If the transition from MSA to LSA is regarded as a cultural analogue of cladogenesis – that is, if there is a branching process whereby the MSA persists after the LSA appears – then the distinction is real, and the examination of penecontemporaneous MSA and LSA assemblages (preferably within the same region) should reveal the key differences. Comparison of long archaeological sequences that span the transition in eastern Africa, such as those at Mumba [3,27], Enkapune ya Muto [2], Nasera [28], Kisese II [71], Lukenya Hill [72], and Panga ya Saidi [1] should prove particularly useful in this context. Of particular interest will be examination of the extent to which sub-regional trajectories are similar in terms of both the tempo and mode of the transition. If sub-regional trajectories are similar in duration but out of phase chronologically it would suggest that sub-regional populations are not in contact with one another, and that the record could be interpreted as indicating convergent but independent trajectories. By contrast, if certain sub-regional trajectories show sudden 'jumps' or accelerated changes at particular dates, these dates may indicate points of contact with other populations that have already accumulated a greater number of technological innovations. This observation is made not to suggest that the transition from MSA to LSA was inevitable, or that it was marked by continuous incremental 'progress', but to suggest that observed differences in sub-regional trajectories may reveal information about the extent of connectivity between populations at this scale.

The analyses above support many of the classical typological indicators of the transition: backed pieces, bipolar reduction, and blades all explicitly indicate the LSA; core tools, Levallois flakes, point technology, and scrapers all either indicate phases of the MSA or contra-indicate the LSA. Levallois blades and points did not reach statistical significance as indicators, and this may suggest the need to subsume these into a single 'Levallois' category in future analyses. There are also potential issues of non-independence: for example, experimental evidence suggests that bipolar reduction may be directly linked to the production of functionally microlithic tools [73,74]. The typology employed here does not distinguish between side- and end-scrapers, which some [e.g. 38] have argued to be another signature of the transition. Backed pieces, bipolar and blade technologies (especially the presence of bladelets) commonly, though not exclusively, focus on smaller tools sizes, which may indicate some interaction with a key change in size of lithic technology associated with the LSA, in contrast the role core tools can play in MSA assemblages. The typology employed does, however, demonstrate considerable power in discriminating between LSA and MSA assemblages, and in delineating changes within the MSA. By using a presence / absence classification, sample size is maximised, and maximum discriminatory power is achieved as a result.

Regardless of one's confidence in the terms 'LSA' and 'MSA', the above analyses demonstrate that their current usage reflects a very real division in the data. The finding that LSA assemblages are indicated by the presence of backed pieces, bipolar technology, and blades, for example, could be re-written as: 'the term 'LSA', as used by archaeologists working in eastern Africa, refers to an industrial complex marked by assemblages that contain backed pieces, bipolar technology, and blades'. The analyses reported above demonstrate that it is possible to distinguish between these polythetic industrial complexes *as they have thus far been employed by archaeologists to describe variability in the lithic record*. Had these analyses failed to find technological differences between assemblages labelled 'LSA' and 'MSA', the natural conclusion would have been that these terms are arbitrary and essentially meaningless. That the 2-way analyses achieve correct classification rates of almost 99% and pick out constellations of technologies that differ significantly in terms of probability of presence between the two industrial complexes suggests that the terms 'LSA' and 'MSA' remain highly valuable.

In summary, the analyses reported above employ neural networks to establish a total of 7 technologies that either distinguish between MSA and LSA assemblages or highlight changes

within the MSA of MIS3-5 in eastern Africa; in doing so, they successfully classify over 94% of assemblages to the correct industry or period. Many of the classical technological indicators of LSA and MSA industries are supported, as is the contention that the MSA to LSA transition was a protracted, complex process. Neural networks are shown to be a valuable tool for archaeological data analysis, and it is suggested that future analyses should focus on further examination of both the typological and the chronological nature of the MSA / LSA transition in terms of tempo and mode, with a particular focus on intra-regional comparison of localiaties with long stratigraphic sequences. Results show that the majority of technologies persist throughout the sequence of MIS2-5 assemblages studied (whether labelled as MSA or LSA), indicating that the transition is marked by key typological changes against a background of considerable continuity. Although there are no *fossiles directeurs* for the LSA or MSA industrial complexes, constellations of technologies aid considerably in classifying assemblages to these complexes. For example, an assemblage containing backed pieces, bipolar technology, and blades, is highly likely to be categorised as LSA. By identifying such constellations of technologies, the analyses provide a clearer notion of what is meant by 'LSA' and 'MSA' as these terms are currently employed by archaeologists.

## Supporting information

**S1 File.**
(DOCX)

**S1 Data.**
(XLSX)

**S1 Code.**
(TXT)

**S2 Code.**
(TXT)

## Acknowledgments

We would like to thank Christian Tryon and four anonymous reviewers for their comments on an earlier version of this paper.

## Author Contributions

**Conceptualization:** Matt Grove, James Blinkhorn.

**Data curation:** James Blinkhorn.

**Formal analysis:** Matt Grove, James Blinkhorn.

**Funding acquisition:** Matt Grove.

**Investigation:** Matt Grove, James Blinkhorn.

**Methodology:** Matt Grove, James Blinkhorn.

**Project administration:** Matt Grove.

**Supervision:** Matt Grove.

**Validation:** Matt Grove, James Blinkhorn.

**Visualization:** Matt Grove.

**Writing – original draft:** Matt Grove, James Blinkhorn.

**Writing – review & editing:** Matt Grove, James Blinkhorn.

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
