## [Decision Letter · Decision Letter 0]

6 May 2020

PONE-D-20-06346

Neural networks differentiate between Middle and Later Stone age lithic assemblages in eastern Africa.

PLOS ONE

Dear Dr Grove,

Thank you for submitting your manuscript to PLOS ONE. After careful consideration, we feel that it has merit but does not fully meet PLOS ONE’s publication criteria as it currently stands. Therefore, we invite you to submit a revised version of the manuscript that addresses the points raised during the review process.

We would appreciate receiving your revised manuscript by Jun 20 2020 11:59PM. To enhance the reproducibility of your results, we recommend that if applicable you deposit your laboratory protocols in protocols.io, where a protocol can be assigned its own identifier (DOI) such that it can be cited independently in the future. For instructions see: http://journals.plos.org/plosone/s/submission-guidelines#loc-laboratory-protocols

We look forward to receiving your revised manuscript.

Kind regards,

Justin W. Adams, Ph.D.

Academic Editor

PLOS ONE

Additional Editor Comments:

Thank you for your patience regarding the review process for your submission with PLoS One. As you can appreciate this is relatively complex time for many researchers, and equally your submission represents a novel blend of statistical/quantitative archaeological analysis within the discipline. Hence the broad number of reviews sought. The consensus view is that while the manuscript satisfies the publication criteria for PLoS One, that there are several areas where improvements could be made to both the overall structure of the publication (through structuring, editing and consolidation) and to the underlying approach towards handling the archaeological data (e.g., categorisation/coding/identification of tool types). I do not see any unwarranted comments or critiques of the manuscript in its current form arising from the reviewers, and in particular, reviewers 2 and 5 have raised a number of fundamental points which (while certainly addressable) will require particular attention to address within any revised submission. The nature of these comments and concerns raised (e.g., as fundamental to the units of analysis) may require a secondary review of the revised manuscript but this can only be determined on receiving the revised document.

2. In your manuscript, please provide additional information regarding the specimens used in your study. Ensure that you have reported specimen numbers and complete repository information, including museum name and geographic location.

For more information on PLOS ONE's requirements for paleontology and archaeology research, see https://journals.plos.org/plosone/s/submission-guidelines#loc-paleontology-and-archaeology-research.

Reviewers' comments:

Reviewer's Responses to Questions

**Comments to the Author**

1. Is the manuscript technically sound, and do the data support the conclusions?

Reviewer #1: Yes

Reviewer #2: Yes

Reviewer #3: Yes

Reviewer #4: Yes

Reviewer #5: No

2. Has the statistical analysis been performed appropriately and rigorously? 

Reviewer #1: Yes

Reviewer #2: I Don't Know

Reviewer #3: Yes

Reviewer #4: Yes

Reviewer #5: Yes

3. Have the authors made all data underlying the findings in their manuscript fully available?

Reviewer #1: No

Reviewer #2: No

Reviewer #3: Yes

Reviewer #4: Yes

Reviewer #5: Yes

4. Is the manuscript presented in an intelligible fashion and written in standard English?

Reviewer #1: Yes

Reviewer #2: Yes

Reviewer #3: Yes

Reviewer #4: Yes

Reviewer #5: Yes

5. Review Comments to the Author

Reviewer #1: This is a very interesting paper that illustrates the application of artificial neural networks to the study of artifact collections and their classification for periodization purposes. The specific case relates to the transition between Middle to Late Stone Age in eastern Africa.

I found the text to be rigorous in the presentation of the problem and the methodology.

The only suggestion I can give to the authors is to evaluate - if they decide to accept my advice - to condense the text as much as possible. I think it is too long and this certainly compromises the incisiveness that this experience can have.

I also recommend the authors to integrate the conclusions and provide a more detailed summary of the results obtained.

Reviewer #2: This is an interesting paper. I recommend publication with minor revisions. Specifically,

1. There needs to be an explicit statement that this method is NOT a substitute for actual geochronological determinations for Eastern African archaeological sites.

2. Somewhere, perhaps in supplemental materials, there needs to be an explicit discussion about how the authors determined whether particular kinds of stone tools were present or absent in a sample assemblage.

I call these things “minor” because #1 requires a sentence or two, and #2 merely involves writing down criteria for presence/absence determinations in supplemental materials.

As a reviewer, I have to preface my remarks with a statement my qualifications. I know virtually nothing about “neural networks.” The authors do an adequate job of explaining them, but because I think it would be wrong to opine about this subject based on just that briefing, I urge you, the Editor(s), to invite reviewers with relevant expertise. Similarly, the paper needs review by someone well-versed in multivariate statistical methods. I am not such a person. I do know fair bit about stone tools, and about stone tools from Eastern Africa, but others certainly know more than I do.

Age-stages

Eastern Africa boasts the world’s longest-running stone artifact sequence, 3.4 million years and still going. (People still make and use stone tools in Eastern Africa.) That record is unquestionably important for research into long term patterns and processes in human evolution. Starting around the mid-20th Century archaeologists working in Eastern Africa (East Africa proper and the Horn), began arranging stone tool assemblages and industries (groups of assemblages) into age-stages. They did this by importing to the region three Pleistocene age-stages then in wide use among Southern African archaeologists (the Earlier, Middle, and Late/Later Stone Ages) and two from western Eurasia (Neolithic and Iron ages). (This is kind of like making guidebook to the birds of American Midwest by binding together sections of guidebooks to the birds of Canada and Mexico.)

Over the ensuing decades, it has become clear that these age-stages do not work particularly well for organizing Eastern African archaeological variability. Ostensibly “MSA” Levallois cores show up in Iron Age contexts. Later Stone Age microliths (small backed-truncated pieces) pop up in Middle Stone Age contexts. Later Stone Age people made ceramics, and so on. But, it’s not that the Eastern African archeological record is intrinsically messy. Archaeologists working in Southern Africa and their colleagues working in western Eurasia have realized that these age-stages don’t work particularly well in those regions, either. South African hominins were making MSA points at Kathu Pan 500,000 years ago, 300,000 years before the conventionally-recognized start date for the MSA there. Iron Age Israelites were still knapping flint sickle blades 3500 years ago. The problem isn’t the archaeological record, the problem is archaeologists’ use of age-stages to organize the archaeological record.

Whenever and wherever archaeologists use age-stages, doing so creates the illusion of homogeneity among the data and the expectation of that homogeneity among archaeologists. The worst outcome of all this is that many archeologists working in Eastern Africa think that they can “date” archaeological assemblages based on those assemblages technological and typological characteristics.

I am concerned that some colleagues, having read this paper, will think that they can now plug their data into this neural network and get a “date” for otherwise undatable assemblages.

(One of the world’s foremost authorities on the geochronology of Eastern Africa, the late Frank Brown, routinely referenced such lithic “dates” using a barnyard epithet.) Peer-review will disabuse them of that idea pretty fast. But, if you propose something that seems to offer a shortcut to knowledge otherwise difficult to obtain, in this case geochronology, your colleagues will use it and then malign you when it fails. I strongly urge the authors to insert in the text a statement warning colleagues to NOT use this approach as the sole basis for dating archaeological assemblages. Colleagues will still try to do this, of course, but the authors will at least be able to point to a specific passage where they warned against doing so. (I write here from prior personal experience with a measurement I proposed that colleagues misused as a “shortcut.”)

Artifact-type identifications.

A house is only as strong as its foundation. This paper’s “foundation,” if you will, are identifications of specific stone artifact-types from published literature. All archaeological artifact and ecofact identifications depend heavily on visually assessed morphological analogy (i.e., “This looks like that”). Stone tools are a little more problematic than most archaeological evidence, because no two stone artifacts are identical in all respects, and because different research traditions diverge in terms of how they describe stone tools. As a result, whenever you get a bunch of archaeologists together and put stone tools in front of them, the conversation goes more or less as follows:

Bob: This is a microlith.

Joe: You call that a microlith! That’s not a microlith!

Bob: Yes it is!

Joe: No it isn’t!

Susan: Colleagues, what do we mean by “microlith”?

Bob: What the late Prof. Smith called a microlith.

Joe: What the late Prof. Jones called a microlith.

Carol: That’s not what I call a microlith.

And so on….

I truly sympathize with the authors. I recently completed a literature-based survey of several hundred African stone tool assemblages using methods much like those on display here. Like the authors, I decided presence/absence was the way to go. Why? To minimize inaccuracy. The more I read, the clearer it became that published artifact-type identifications were made by different archaeologists working decades apart, in different research traditions, with differing degrees of familiarity with lithic technology, using different terms for the same things and the same terms for different things. Sometimes text and artifact illustrations correlated with one another, but often they did not. One assumed artifacts illustrated were accurately drawn and portrayed representative samples, even though one knows this varied and varies. One took on faith that individual scholars described stone tools consistently throughout their professional careers, but no evidence whatsoever supports that hypothesis. Inter-analyst identification variation is a problem in every region of the world; but, because so many archaeologists from different research traditions work in Eastern Africa and because scholars there all-too-often hand over the describing of stone tools to graduate students with little (literally, the least) experience in stone tool analysis, Eastern Africa may actually be WORSE in these respects than many other regions. I could not find a single paper in which different scholars published and compared their individual assessments of the same set of artifacts and then discussed how to reconcile differences. I found only one paper (Mike Mehlman’s unpublished dissertation) that even tried to establish concordances between different archaeologists’ stone tool systematics.

Terminological variability is not just a problem with the older literature, much of which archaeologists created working independently, before the Internet and digital photography made friction-free comparisons possible. Junior colleagues researching the African Middle Stone age recently met at Harvard to try to correlate their stone tool systematics. They met for 3 or 4 days and apparently could finally only agree on how to measure flake length. My point, this is not just a problem with the older literature, the seeds of Eastern Africa’s lithics systematics anarchy have sprouted and are growing fast. Soon too, they will bear fruit.

To firm up this neural-net approach’s “foundation, the paper could benefit from supplemental materials definining of the specific artifact types (with emphasis on the more diagnostic of these) as well as detailed notes on how the authors made present/absent determinations from reading the literature. Doing this will increase the probability that the findings published in this paper will be replicable. Just to be clear, this needs to tell how to recognize the artifacts themselves AS WELL AS how to recognize their occurrence in published literature. They don’t need to make some kind of massive concordance among the typologies everyone has used in Eastern Africa from O’Brien and Leakey up to the present. Perhaps just list of key term and common synonyms, for example “Still Bay point” (Author 1 [1955], aka “foliate point” of Author 2 [1981]).

Some miscellaneous suggestions follow:

Bipolar technology. There are between 6 and 8 different terms currently in use for pieces of stone bearing fractures from having been set on one stone and struck forcefully from above with another one (see scaled pieces, below). One needs to clarify which terms you recognize as bipolar technology.

Blade technology. Probably a good idea to reproduce the conventional length/width definition here and such other criteria as seem reasonable (parallel lateral edges, parallel distal-proximal dorsal flake scars, etc.).

Discoidal cores: Need to make clear these are different from radial/centripetal cores. Many researchers do not make the distinction clearly.

Large cutting tools. So, basically handaxes, picks, core-axes, and lanceolates. These things aren’t particularly large. On average, the handaxes are about the same size and mass as an iPhone. Only a tiny number of them actually preserve wear traces from cutting. Consider calling them “long core-tools” (same acronym, LCT). One might also specify a size threshold or width/thickness criterion. MSA “handaxes” grade into “foliate points” and thence downwards in size to stubbly little bifacial cores that are sometimes called points, other times different terms such as “elongated discoids”.

Levallois flakes. In older literature, use of the term Levallois pretty straightforwardly tracked Francophone research, but this is less a problem among recent works (many researchers having adopted Boëda’s criteria for recognizing this). One has to be particularly alert that conventions for naming stone tools “Levallois” differ sharply between reports on MSA vs. LSA that this almost certainly reflects archaeological naming conventions for research in different time periods rather than actual changes in stone tool typology. (One sees this in many other regions as well. Levallois cores become “discoidal cores” or something else when they turn up in post Middle Paleolithic/Middle Stone Age occurrences.)

Microliths. Definitions of these things are all over the place. One should specify at least a range of length thresholds for identifying them. Most fall between 30-50 mm, but I have seen artifacts as long as 10mm called microliths -especially in MSA contexts. (As the text notes, microliths are not the same thing as miniaturized stone tools. Such tools can take different forms (e.g., small Levallois flakes and cores) from small backed/truncated pieces.

Points. Points are a mess -a true garbage can taxon. One needs to make clear whether one is including unretouched pointed flakes, points with a retouched tip and/or basal thinning, or bifacially-worked pieces, or if instead one is casting the net wide and including every medio-laterally symmetrical piece with a retouched distal end.

Radial cores. Recurrent radial/centripetal “Levallois” cores and distinctive flakes (“pseudo-Levallois points”) get treated differently by different researchers. Some consider these things Levallois, others do not. Some distinguish hierarchical discoidal cores (longer flake scars on one side of core/worked edge) from radial cores (non-hierarchical discoidal cores -similar flake scars on different sides of the edge), other do not make this distinction (particularly people working on LSA samples).

Scaled pieces. Many of these things are just flakes used as bipolar cores. Again, terms for them vary. I do not see a strong reason to discriminate them from bipolar cores.

To sum up, this is a good study and an impressive demonstration of this neural net approach to assemblage classification. I think people will try to use it to “date” undated assemblages no matter what you do, but one should warn them against doing so in the text.

I am definitely NOT recommending the authors undertake an overhaul of Eastern African MSA and LSA stone tool typology. I am recommending they add supplemental materials that will aid colleagues in replicating their approach with additional assemblages.

Reviewer #3: Grove and Blinkhorn present a well detailed and easy to follow original research article. The experiments, statistics, and analyses are performed to a high technical standard and each are described to a level of detail that allows the reader to follow the chain of logic behind the authors' conclusions. Data is present in a clear fashion and all data underlying the findings available in the supporting information files, within the body of the text or within the figures. The manuscript provides a logical succession to their 2018 paper (The structure of the Middle Stone Age of eastern Africa), utilising the same broad stone tool typology, allowing for evaluation of variability in behaviours and assemblage compositions both within the MSA and the changes that take place moving into the LSA.

Some small type edits should be addressed:

• Line 158 change Fig 1 to Figure 1, for consistency within text

• Line 580 change (see Figures 4 and 5) to (see Fig 4 and Fig 5), for consistency within text

Reviewer #4: Review of Grove and Blinkhorn 2020 PLoS One

This is an interesting and potentially very important paper, both in terms of developing a novel methodology and for providing some rather robust data for describing the actual process of typo-technological change in lithic technology across the Middle/Later Stone Age transition, an important period of archaeological change likely connected in some way to the dispersal of Homo sapiens across and out of Africa. My comments below are very much written in the spirit of wanting to make the published version of this paper as clear, strong, and impactful as possible, and they should be read in that way. I definitely recommend publication after revisions and further review. My suggested changes straddle the minor/major revisions category, but because I think they require at least one more set of analyses, I’m recommending major revisions as the more conservative option. I don’t normally like to sign my reviews, but I will here (this is Christian Tryon) in case the authors wish to contact me about a few points; there are some arcane details about East African lithic typology that I might be able to help them with. That is, I have spent most of my career wrestling with some of the problems dealt with in this paper; Grove and Blinkhorn have now worked in East Africa for quite some time but received initial training in other regions. I actually think that this makes their ideas far more interesting than mine, but there are some important details about the region (and its intellectual history) that I might be able to assist with. Unfortunately, too much of my brain is cluttered with these sorts of trivia.

I’ve presented my comments below mostly in the order that they came up in while reading the manuscript; I’m sorry for not being able to rewrite these in a more synthetic fashion, but being homebound with kids and a working partner leaves me with considerably less time for academic work than usual.

Line 26 and elsewhere: The authors go back and forth on their use of Late or Later Stone Age. I suppose that I don’t really care, but if we’re going back to the original Van Riet Lowe and Goodwin terms, then it should be “Later.” Whatever the choice, it should be consistent throughout.

The introduction: I absolutely do not want to come across as the “why didn’t you cite my work?” guy, but I did just publish a review about the MSA/LSA transition in East Africa a few months ago in Evolutionary Anthropology, and it would seem reasonable to mention this in the introduction. However, I am not suggesting that they simply cite this and cut out stuff; I like how the authors set up the problem and outline what we already know. If anything, I’m suggesting that they use my paper as a foil. One of the things that I did not do, even though the reviewers asked me to do it, was to provide a solid definition of how one knows whether or not a site is “MSA” or “LSA.” I resisted this because I didn’t feel that I had a solid basis on which to make this sort of definition. I think that the paper by Grove and Blinkhorn actually does this, which is part of the reason why I think it’s a great paper. If anything, my work could be cited in the context of “even recent reviews fail to actually provide a solid, workable definition of what the terms MSA and LSA mean” or something like that. Building on that, I do think that the paper would benefit a little bit from some more reflection on whether or not terms like MSA and LSA are even useful. Certainly, some, like John Shea, would say absolutely not. Others, like Foley and Lahr and Barham and Mitchell, have pointed out problems with them as well and defaulted to Clark’s “modes” as an alternative. I don’t really feel strongly either way (for me they are useful but flawed terms), but I do think that situating the reader a bit more into why these terms are potentially useful but so poorly defined would be a good thing. I worry that otherwise it might not be obvious why this sort of analysis is in fact quite interesting.

Line 83 or so: I’m not a big fan of the MIS stages personally, but I do understand that they can be useful “boxes” when aggregating data, especially when the chronological resolution of most sites is poor. But because I don’t use them, I can never remember when the hell the boundaries are for these time periods. Can the authors include their chosen time boundaries for the MIS stages throughout the text? I know that it’s a small thing but again I do think it’s helpful because it makes the results a bit less opaque, and (I hope!) I’m probably not the only person who can never remember the age boundaries.

Line 91 or so: This is the paragraph where I would expand a bit on what the MSA and LSA categories refer to (see above also): taxonomic identifications that were originally a stand-in for stratigraphy and chronology from the early days of the discipline. I would note that there is a history of attempts to abandon them (Barham and Mitchell review this), but that the current paper succeeds by offering a good definition based on the most abundant type of artifact: stone tools and lithic debris (I stress this last bit because the rare but fancy personal ornaments and worked bone get more press, but from a numbers game the stone tool data should win every time, and I think that few people actually appreciate this!).

Line 147 or so: I am completely sympathetic to this issue of poorly defined terms. Some of these issues are made explicit in the 2019 Evolutionary Anthropology paper, and also in a small conference review (Will et al in Evolutionary Anthropology). The latter need not be cited, but it does at least show that several of us are actively working on figuring out the mess that is African lithic technology. Another useful thing to reference here would be John Shea’s new book (I think it was published last week) on East African lithic typology, where he outlines some of these problems as well.

Methods, data: This is one of my biggest problems with the paper. I don’t really understand which data were used and which data were not used. The data table in the SI has several spelling errors, and no citations are given from where the data actually derive. I think this is a problem. I understand that some of these data were published in previous paper (Blinkhorn and Grove 2018), but this is not the case for the LSA data, which are new to this paper. Also, it is made clear ONLY in the discussion that the MSA or LSA attribution in the table is made based on the original publication, and not necessarily on the authors’ assessment. This is REALLY important! I couldn’t tell what was actually being tested until the paper was almost over (this is mentioned in the discussion section only!). Also, I think, but could not tell, if the sites in the SI table are only those that were used as test assemblages and not as training assemblages. If correct, this needs to made very clear; the implication is that there is a huge training set with data not included in the SI, which would make it hard for anyone to replicate the analyses here, I believe, and it makes it hard for me to judge the reliability of the results (see below about blades). I also think that a map of the sites discussed should be included. My aim here is to simply make it as clear to the reader as possible and which data are being used what’s going on because I really want this paper to succeed; it shouldn’t be necessary to read between the lines or to have to dig through an older publication just to work out what is actually being analyzed. Hopefully this is a pretty quick fix.

Line 253 or thereabouts: See comments above about needing to be clear about who made MSA or LSA attribution. My original comment here was something like “why don’t you just try by age instead?” a question that was addressed (and addressed quite reasonably!) in the discussion section. I think that it would be good to move up the rationale behind why the analyses were done earlier on in the paper.

Line 332: Just to clarify; the sample size of 10 is considered sufficient by the authors, or by Baxt? That is, is this a true generalization or something specific to these analyses or this dataset? I appreciate that this paragraph is written in a very digestible way, but I think that making this point clear is important for future users.

Line 348: This should read "greater than or equal to" for the example to work, right?

Line 424: There is a VERY real need to define your terms here, especially what is meant by “East Africa” and why some sites are included but not others. For example, why aren't any of the sites at Gademotta/Kulkuletti included, but other Ethiopian sites are? Again, is it because those sites are in the training but not the test assemblage? The lack of early blades and the absence of sites from Gadetmotta/Kulkuletti just really threw me because I'm pretty sure those sites have blades and I'm definitely sure that they're early, and as there is a published monograph (or two, if you count Katja Douze's theses) on the site, the data should be available. Anyway, I may be wrong about this particular site, but I do feel that some definition of terms (space ,time, and artifacts) are needed here. I appreciate that some of these things are spelled out in an earlier paper, but a reader shouldn't need to read an earlier paper just to understand the data set in this one, in my opinion. The same goes with the table; what is an RTBifacial, for example? I dug through the Blinkhorn and Grove (2018) paper and figured out that it’s a retouched bifacial piece, but those definitions should all be included in this paper as well, at least in the SI if nowhere else.

Discoidal and radial cores: This is the big problem with an obscure bit of terminology. Discoidal cores and radial cores are often the same thing. I knew that there was a problem when I read the statement that discoidal cores are rare. They’re everywhere, but they go by such a large number of names that it gets really confusing. The term ‘discoid’ was used by Mary Leakey for Olduvai Gorge, and sometimes that terms gets used for MSA assemblages, but most researchers in the 1960s-1980s working in East Africa used the term radial core to mean the same thing (something flaked bifacially about the periphery towards the center of the piece). Influenced by those working outside the region (Kuhn, Tixier, Boeda), the terms ‘centripetal’ and ‘discoidal’ crept in in the 1990s. It was made even more confusing by the Kalambo Falls Volume 3 monograph, which grouped discoidal cores and Levallois cores as ‘prepared’ cores. There may some variation between users, but I am sure that the existing division between discoidal and radial cores has caused a real problem in the dataset. I actually think it could probably be sorted out pretty quickly (and this is something that I could help with), but it would require the analyses to be re-run. I would be VERY surprised if the final results differed much, but I do think that it’s an important thing to sort out before publication. The best places to see the overlap in these terms is to compare the typology chapters in the theses of Harry Merrick and Mike Mehlman.

Line 527: This may or may not be important, but ‘LSA’ is not usually referred to as an ‘industry’ but something higher up in the taxonomy, like ‘industrial complex’ or something like that (see Clark et al 1966 in South African Archaeological Bulletin). Not a big deal, really.

Line 596 and thereabouts: I find these results super exciting, especially since I’ve worked on nearly all of the assemblages that get misclassified! I think that the authors should really take the chance here or elsewhere to really emphasize that their ‘misclassifications’ actually make sense and support a couple of existing hypotheses. The first is the level 4/5 stuff from Nasera. You have to kind of read closely because we hedged things a bit, but see page 8 on Ranhorn and Tryon (2018) in the Journal of African Archaeology. We re-dated levels above and below level 4 at Nasera because we didn’t have any samples from level 4 (we’ve since found some and are dating them now). But based on the radiocarbon dates published in that 2018 paper and older AAR dates that we cite that are actually probably correct, the age of Nasera 4/5 is much older than previously thought and perhaps closer to the MIS 3 or 4 boundary (see, I had to look up the age boundary just now!) and probably equivalent in age with Mumba Bed V (one of the other problem assemblages). Therefore, the misclassification actually supports the idea laid out in Ranhorn and Tryon, which is great (that level is older than it was previously believed). The other site that’s a problem, Karungu, is also one that I’ve been involved with, and that it classifies as being older than it is also supports a couple of existing observations. Blegen et al (2017 in Paleoanthropology) stress that the Lake Victoria basin has a lot of ‘young’ MSA sites compared to surrounding regions, and in my 2019 Evolutionary Anthropology paper I also stress that the fauna and absence of OES beads from that region suggest at best weak connections to surrounding areas. That is, there are other lines of evidence that suggest that the Lake Victoria region shows a persistence of ‘older’ technologies relative to areas around it. I guess my bigger point is that the existing text focuses on the methodological issues (e.g., marginal classification values), which are great. But that it seems like a good opportunity to ALSO consider how the results mesh with other lines of evidence.

Lines 675-678: Can you provide a basic table that would give predictive power for the classification of future assemblages, if the MSA/LSA terminology is deemed useful? Or is this approach only backward looking? That is, as exciting as the methodological development is, do the results now allow us to move forward with increased confidence in what we call a site, or does it only work for older, already published data? I am of two minds on this issue, but I do think that demonstrating predictive power for new assemblages would increase the future importance of this paper.

Anyway, I really enjoyed reading this paper, and I sincerely hope that my comments are useful in some way; they are certainly meant to sharpen the arguments of a very interesting piece of work.

Reviewer #5: I found this article interesting and believe it ought to be published, but that it needs significant addition, restructuring, and correction before it can be accepted for publication. My own experience is in the archaeological, rather than statistical/mathematical aspects of this research, and my comments therefore focus on this aspect of the paper. Regarding review questions 2 and 3 (is statistical analysis appropriate and rigorous / is data underlying the findings made fully available) I have no substantial concerns, however some of my comments outlined below do impinge on these issues. Similarly, the paper is written in an intelligible fashion and in standard English (Review question 4), but does require some substantial re-structuring (see comments below).

My concerns are principally in the following areas relating to Review Question 1, and which I consider require substantial revision and modification: a) the theoretical positioning of the paper and question formulation in relation to its conclusions is circular. The discussions/conclusions of the paper do not match what I would assume are the most salient outcomes of the analysis. b) critical analysis of legacy archaeological data sources used for statistical analysis (including their assemblage descriptive statistics, date of excavation etc), and sincere discussion of the integrity and inter-comparability of these independent datasets is completely absent. A discussion of this should be foregrounded and include how and why data was sampled (selection) and issues such as whether the authors have assumed/verified typological nomenclature is uniformly applied to assemblages within and between sites. The potential problems inherent in using typological data for this kind of analysis must be included, and; c) confusion throughout between the concepts of typology and technology, and the interchangeable use of these non-overlapping terms, which are foundation to the work. In addition, I outline some more minor points needing attention at the end of my review.

a) The paper appears positioned to do two things concurrently – demonstrate the efficacy of the ANN statistical methodology and also resolve the major differences between the MSA and LSA on either side of this technological transition, as identified and described previously by archaeologists. The paper seems, however, to be focused in neither arena, to the detriment of both.

Any ‘test’ of ANN cannot be reasonably undertaken on data that – at whatever level one wishes to consider it – is not objectively categorical (i.e. some assemblages appearing more LSA may also have strong commonalities or chronological affiliations with the late MSA, or vice versa): for such a graduated transition, affiliation to the MSA or LSA as determined by archaeologists is themselves somewhat debateable. Based on the statement (included very late in the article) that ‘the database employed here accepts the excavator’s assessment of industrial affiliation in all cases’, it appears the authors benchmark of success is whether it agrees with the qualitative interpretations of the archaeological record made previously by archaeologists. This is barely stated, but the paper is therefore testing the accuracy of the ANN against a subjective benchmark. The article must directy address this issue in the text before the analysis is presented.

It is stated in the abstract and conclusions that the methodology accurately separates assemblages in up to 95% of cases, however this rate is only achieved when particular artefact types in the assemblages are removed from the statistical analyses, and therefore its actual rate of accuracy, as directly applied to all tested tool types in the assemblages, is lower than 95%. The point is: the claimed rate reflects neither the success of the ANN methodology (which requires some modification to achieve this degree of accuracy on these assemblages) nor a real-world measure of the separability of the archaeological assemblages based on their entire techno-typological content. The paper needs to be modified so that the discussion and conclusions address one or other of these issues, until which time it cannot claim to achieve either of these aims.

There are a few ways these related problems could be addressed: re-writing the paper to make abundantly clear that this rate of ‘success’ is contingent on selected assemblage components and that the ‘benchmark’ of ‘success’ is considered to be when the ANN methodology classifies assemblages into the same temporal/technological categories as determined by the excavators (i.e. MSA/LSA), or; repositioning the paper and methodology as an objective method for separating these assemblages into early/late MSA / LSA categories, which the authors consider to be more objective – and ergo more accurate – than the results achieved by qualitative lithic analysis by archaeolosgists. This latter solution would seem at odds, however, with the potentially problematic data upon which the analysis is driven (see point b, below).

A final point I find problematic is the circular nature of the paper’s logic: the authors state clearly and correctly that the MSA-LSA transition is nuanced and about degrees of presence/absence (scale data) rather than outright categorical change (Line 102). The attempt is then made to examine the presence/absence of artefact types and technologies to categorise the differences between pre and post-transition assemblages, with the finding being that the chrono-technological units (earlier/later MSA/LSA) cannot be easily discretized because their differences in techno-typology are nuanced and graduated rather than categorical. Isn’t this therefore simply a quantified reconfirmation of what archaeologists have already recognised through lithic analyses? Wouldn’t a statistical methodology assessing proportional change in lithic data in fact be a better tool given what is already known about the mosaic nature of the technological changes between the MSA and LSA? This kind of confirmatory conclusion would be appropriate (and very interesting) if directly expressed as a key focus of the paper and data analysis (which would also fit with a focus on a test of the ANN method), but as it stands, I’m personally unsure how the paper enhances our understanding beyond what has already been recognised through standard archaeological analyses of the assemblages.

b) As presented, the analysis hinges on the use of legacy (particularly typological) data extracted from publications of East African excavations. However, there is no serious description of how or why these assemblages have been selected, or an interrogation of these datasets to establish the suitability of the categorical data (typology) for comparative statistical analysis. Moreover, there is only cursory mention of the fact that any typology, as a snapshot output of dynamic technological systems that are conditioned by multiple independent factors, is a relatively blunt tool for assessing change. I am deeply concerned by these omissions as they potentially undermine the conclusions of the paper entirely. The uncritical and incautious utilisation of data from disparate sources may actually serve to generate new, inaccurate archaeological inferences, rather than to clarify existing problems.

It is mentioned (Line 60) that the ‘number of published eastern African assemblages is now sufficient [for analysis]’, but there is no description of why the particular sites and assemblages selected have been chosen (availability of published data?), the chronological and historical context of their excavation, any divergences in the typological schemes applied to the assemblages, the samples’ size or integrity, or their representativeness (especially in light of the various factors including raw material availability, distance, type and form that can affect lithic reductions strategies and the ‘types’ that result). These are key issues at the level of the fundamental data input that must be addressed before the output of the statistical analysis can be considered meaningful. The method of sampling used for statistical analysis is always important if one is to appropriately understand the limits of the data output. The lithic assemblages used here are sub-samples (reported information, in compressed, typological format), of samples (the excavated material) of an original sample (% of site excavated) and as such are highly selected before being any site-site comparison. Demonstrating an understanding of nuance of the original data is fundamentally important in synthetic statistical studies such as this, as it helps mitigate disconnects between source data units (artefacts) and statistical outputs. A detailed description of the sampling strategy used and justification and potential problems with comparing typological data directly between sites should be included.

Relating to sample size, there is inconsistency in the interpretation of ‘significant tool types’ when those types are low in frequency. An example is between LCTs and ‘Point technology’ (nb: points are a type, produced through levallois (or PCT) reduction – see pont c, below). It is stated that points are found co-present with ‘stronger indicators of MIS4&5 MSA assemblages such as LCTs and notches’, but in the section on LCTs their numbers and proportionate presence in and between the LSA and MSA assemblages is almost identical. How then can this inference be substantiated? There are similar examples of such apparent contradictions in interpretation, giving a sense that the inferences drawn are not robust and are therefore difficult to trust.

Striking is the fact that – so far as I can see – at no point do the authors state directly that they have taken as read the typological assessments made by the various excavators of the material. I assume this is the case, however. The problem with this kind of approach, performed uncritically and without assessing the intercomparability of the typological schemes applied to the various assemblages is highlighted by a case the authors themselves raise. Leplongeon’s work (Line 536) shows there is a key but subtle distinction between the observational units (artefacts) and terms applied. There is rarely consistency in the application of typological names since they inherently compress subtleties in technological information and their application varies across time and space, and between observers. Assessment of the inter-comparability of the assemblages in this study could be achieved by viewing the collections, or examining how typological terms have been applied by the different excavators using the source publications, and relating terms to any illustrations provided. Such a step should be considered critical. Where relevant, this ought also to include a consideration of how lithic nomenclature has changed over time between the publication of the data. Without assessing or addressing this issue, how can the authors be reasonably sure that the statistical analyses applied are drawing substantive and appropriate behavioural inference from the analysis?

Moreover, though it is stated in relation to artefact dimensions that ‘their means of reporting vary more considerably than artefact typologies.’, a more substantive discussion of the theoretical problems in harnessing primarily typological data should be included. I note that the authors apparently ‘acknowledge issues relating to reduction of lithic assemblages’ (Line 147), but there is no serious discussion of the fact that any assemblages merely captures an arbitrary point in the reduction of lithics: the final range of typological components is not therefore fixed or static but an outcome of dynamic behaviour. How might this have limited the output and interpretation of the resulting patterns? This requires detailed discussion and consideration.

c) Finally, for a research paper relying upon the key concepts and accurate data collation of typological and technological information, I am alarmed by the consistent confusion throughout the paper between these (and other) key terms, which are seemingly considered by the authors to be interchangeable. They are absolutely not. The authors, for example, state ‘…key typological components, including...bipolar technology…blade production…and backed geometric pieces’ (Line 96); but none of these are typological components; they are technologies. Similarly, the authors refer to ‘assemblage types’ (Line 521), by which it is seemingly that meant MSA and LSA assemblages can be considered ‘types’, but this is also incorrect. The first section under ‘significant tool types’ is ‘blade technology’, but this and many other listed data categories are not tool ‘types’ at all. Perhaps ‘significant lithic components’ or similar, could be used instead. The interchangeable use of all of these terms runs throughout the paper. Revision of language to ensure correct use of terms applied to refer to the sub-categories of the three-age system, industries, assemblages, types, technologies, etc - none of which are interchangeable - should be undertaken wholesale for the manuscript.

This may seem pedantic, but it’s an important point. Fundamentally, the results of any study hoping to analysing patterns in archaeological data that the authors have not themselves examined first-hand involves the authors demonstrating they understand the nuances of the original source data before running statistics on it: the correct useage of well-defined terms is incredibly important in establishing confidence that that the results are meaningful and robust.

Minor issues:

Line 125 – composite hafting. All hafting is by definition composite tool manufacture and vice versa. The use of these synonymous terms is here redundant and should be amended.

Line 127: ‘emerging from the turn of the century onwards’. This body of evidence in fact began emerging long before McBrearty and Brooks’ paper in 2000; as long ago as 1988 when Clark’s seminal paper on regional identity in the African MSA was published. This historical-contextual comment should be revised to reflect that this evidence base was already well established – and growing - long before 2000.

Line 253: ‘Earlier MSA’. None of the analysed assemblages can be considered ‘Early’ or ‘Earlier’ MSA, which would entail assemblages from MIS 7 or before. I assume the authors mean ‘the older of our MSA samples’. I suggest revision to ‘Due to the unequal size of our LSA, MIS3 MSA and older MSA samples….’, or similar.

Line 414: ‘for both addition and removal and for each…’ needs revision for grammar.

Line 641: ‘the results presented above suggest that this estimate should certainly be viewed as a minimum’: this point needs significant expansion and discussion. As currently formulated, it suggests the authors consider MSA-LSA technological change as a one-way linear process that could not have occurred rapidly. Why should this not be the case? Evidence from the Late Pleistocene lithic record of sub-Saharan African includes key examples of apparently very rapid technological change that contradicts this assumption. This requires reformulation and clarification, or redaction.

Finally, I would recommend revision of the word ‘simply’ throughout the manuscript: it is used multiple times in the same context, often in quick succession, and is superfluous.

6. PLOS authors have the option to publish the peer review history of their article (what does this mean?). If published, this will include your full peer review and any attached files.

Reviewer #1: No

Reviewer #2: No

Reviewer #3: No

Reviewer #4: Yes: Christian A. Tryon

Reviewer #5: No

---

## [Author Response · Author response to Decision Letter 0]

29 Jun 2020

We have uploaded a 'Response to Reviewers' file, a copy of the revised manuscript with changes indicated, and a copy of the manuscript without these changes indicated.

---

## [Editor Report · Decision Letter 1]

29 Jul 2020

Neural networks differentiate between Middle and Later Stone age lithic assemblages in eastern Africa.

PONE-D-20-06346R1

Dear Dr. Grove,

We’re pleased to inform you that your manuscript has been judged scientifically suitable for publication and will be formally accepted for publication once it meets all outstanding technical requirements.

Kind regards,

Justin W. Adams, Ph.D.

Academic Editor

PLOS ONE

Additional Editor Comments (optional):

Thank you for providing your revised submission and your careful consideration of the reviews of your original manuscript. I have gone through your response to the reviewers and the marked revised submission, and appreciate your efforts to address the comments by all the reviewers (particularly Reviewers 2, 4 and 5). While I recognise (as you highlight) that there was somewhat conflicting advice between the message to condense the submission (Reviewer 1) in contrast to requests to expand discussion points or typological definitions, I believe you have threaded the needle appropriately. I particularly note the development of the new Supplementary Information section to help clarify the rather complex historical/current tool technology terminologies. I appreciate that this has required expansion of the manuscript overall, but I do believe it has led to greater clarity. I am happy to recommend acceptance of the manuscript for publication without additional peer-review of the revised manuscript.
---

## [Editor Report · Acceptance letter]

31 Jul 2020

PONE-D-20-06346R1 

Neural networks differentiate between Middle and Later Stone age lithic assemblages in eastern Africa. 

Dear Dr. Grove:

I'm pleased to inform you that your manuscript has been deemed suitable for publication in PLOS ONE. Congratulations! Your manuscript is now with our production department. 

Kind regards, 

on behalf of

Dr. Justin W. Adams 

Academic Editor

PLOS ONE